# An Efficient Tester-Learner for Halfspaces

**Aravind Gollakota**[*] **Adam R. Klivans**[†] **Konstantinos Stavropoulos**[‡] **Arsen Vasilyan**[§]
Apple         UT Austin       UT Austin          MIT

## Abstract

We give the first efficient algorithm for learning halfspaces in the testable learning model recently defined by Rubinfeld and Vasilyan (RV23). In this model, a learner certifies that the accuracy of its output hypothesis is near optimal whenever the training set passes an associated test, and training sets drawn from some target distribution must pass the test. This model is more challenging than distribution-specific agnostic or Massart noise models where the learner is allowed to fail arbitrarily if the distributional assumption does not hold. We consider the setting where the target distribution is the standard Gaussian in $d$ dimensions and the label noise is either Massart or adversarial (agnostic). For Massart noise, our tester-learner runs in polynomial time and outputs a hypothesis with (information-theoretically optimal) error $\mathrm{opt}+\epsilon$ (and extends to any fixed strongly log-concave target distribution). For adversarial noise, our tester-learner obtains error $O(\mathrm{opt}) + \epsilon$ in polynomial time. Prior work on testable learning ignores the labels in the training set and checks that the empirical moments of the covariates are close to the moments of the base distribution. Here we develop new tests of independent interest that make critical use of the labels and combine them with the moment-matching approach of (GKK23). This enables us to implement a testable variant of the algorithm of (DKTZ20a; DKTZ20b) for learning noisy halfspaces using nonconvex SGD.

## 1 Introduction

Learning halfspaces in the presence of noise is one of the most basic and well-studied problems in computational learning theory. A large body of work has obtained results for this problem under a variety of different noise models and distributional assumptions (see e.g. (BH21) for a survey). A major issue with common distributional assumptions such as Gaussianity, however, is that they can be hard or impossible to verify in the absence of any prior information.

The recently defined model of testable learning (RV23) addresses this issue by replacing such assumptions with efficiently testable ones. In this model, the learner is required to work with an arbitrary input distribution $D_{\mathcal{XY}}$ and verify any assumptions it needs to succeed. It may choose to reject a given training set, if it accepts, it is required to output a hypothesis with error close to $\mathrm{opt}(\mathcal{C}, D_{\mathcal{XY}})$, the optimal error achievable over $D_{\mathcal{XY}}$ by any function in a concept class $\mathcal{C}$. Further, whenever the training set is drawn from a distribution $D_{\mathcal{XY}}$ whose marginal is truly a well-behaved target distribution $D^*$ (such as the standard Gaussian), the algorithm is required to accept with high probability. Such an algorithm, or tester-learner, is then said to testably learn $\mathcal{C}$ with respect to target marginal $D^*$. (See Definition 2.1.) Note that unlike ordinary distribution-specific agnostic learners, a tester-learner must take some nontrivial action *regardless* of the input distribution.

The work of (RV23; GKK23) established foundational algorithmic and statistical results for this model and showed that testable learning is in general provably harder than ordinary distribution-specific agnostic learning. As one of their main algorithmic results, they showed tester-learners for the class of halfspaces over $\mathbb{R}^d$ that succeed whenever the target marginal is Gaussian (or one of a more general class of distributions), achieving error $\mathrm{opt} + \epsilon$ in time and sample complexity

[*]`aravindg@cs.utexas.edu`
[†]`klivans@cs.utexas.edu`
[‡]`kstavrop@cs.utexas.edu`
[§]`vasilyan@mit.edu`

$d^{\widetilde{O}(1/\epsilon^2)}$. This matches the running time of ordinary distribution-specific agnostic learning of half-spaces over the Gaussian using the standard approach of (KKMS08). Their testers are simple and label-oblivious, and are based on checking whether the low-degree empirical moments of the unknown marginal match those of the target $D^*$.

These works essentially resolve the question of designing tester-learners achieving error $\mathsf{opt} + \epsilon$ for halfspaces, matching known hardness results for (ordinary) agnostic learning (GGK20; DKZ20; DKPZ21). Their running time, however, necessarily scales exponentially in $1/\epsilon$.

A long line of research has sought to obtain more efficient algorithms at the cost of relaxing the optimality guarantee (ABL17; DKS18; DKTZ20a; DKTZ20b). These works give polynomial-time algorithms achieving bounds of the form $\mathsf{opt} + \epsilon$ and $O(\mathsf{opt}) + \epsilon$ for the Massart and agnostic setting respectively under structured distributions (see Section 1.1 for more discussion). The main question we consider here is whether such guarantees can be obtained in the testable learning framework.

**Our contributions.** In this work we design the first tester-learners for halfspaces that run in fully polynomial time in all parameters. We match the optimality guarantees of fully polynomial-time learning algorithms under Gaussian marginals for the Massart noise model (where the labels arise from a halfspace but are flipped by an adversary with probability at most $\eta$) as well as for the agnostic model (where the labels can be completely arbitrary). In fact, for the Massart setting our guarantee holds with respect to any chosen target marginal $D^*$ that is isotropic and strongly log-concave, and the same is true of the agnostic setting albeit with a slightly weaker guarantee.

**Theorem 1.1** (Formally stated as Theorem 4.1). *Let $\mathcal{C}$ be the class of origin-centered halfspaces over $\mathbb{R}^d$, and let $D^*$ be any isotropic strongly log-concave distribution. In the setting where the labels are corrupted with Massart noise at rate at most $\eta < \frac{1}{2}$, $\mathcal{C}$ can be testably learned w.r.t. $D^*$ up to error $\mathsf{opt} + \epsilon$ using $\mathrm{poly}(d, \frac{1}{\epsilon}, \frac{1}{1-2\eta})$ time and sample complexity.*

**Theorem 1.2** (Formally stated as Theorem 5.1). *Let $\mathcal{C}$ be as above. In the adversarial noise or agnostic setting where the labels are completely arbitrary, $\mathcal{C}$ can be testably learned w.r.t. $\mathcal{N}(0, I_d)$ up to error $O(\mathsf{opt}) + \epsilon$ using $\mathrm{poly}(d, \frac{1}{\epsilon})$ time and sample complexity.*

**Our techniques.** The tester-learners we develop are significantly more involved than prior work on testable learning. We build on the nonconvex optimization approach to learning noisy halfspaces due to (DKTZ20a; DKTZ20b) as well as the structural results on fooling functions of halfspaces using moment matching due to (GKK23). Unlike the label-oblivious, global moment tests of (RV23; GKK23), our tests make crucial use of the labels and check *local* properties of the distribution in regions described by certain candidate vectors. These candidates are approximate stationary points of a natural nonconvex surrogate of the 0-1 loss, obtained by running gradient descent. When the distribution is known to be well-behaved, (DKTZ20a; DKTZ20b) showed that any such stationary point is in fact a good solution (for technical reasons we must use a slightly different surrogate loss). Their proof relies crucially on structural geometric properties that hold for these well-behaved distributions, an important one being that the probability mass of any region close to the origin is proportional to its geometric measure.

In the testable learning setting, we must efficiently check this property for candidate solutions. Since these regions may be described as intersections of halfspaces, we may hope to apply the moment-matching framework of (GKK23). Naïvely, however, they only allow us to check in polynomial time that the probability masses of such regions are within an additive constant of what they should be under the target marginal. But we can view these regions as sub-regions of a known band described by our candidate vector. By running moment tests on the distribution *conditioned* on this band and exploiting the full strength of the moment-matching framework, we are able to effectively convert our weak additive approximations to good multiplicative ones. This allows us to argue that our stationary points are indeed good solutions.

**Independent and Subsequent Works.** In this paper we provide the first efficient tester-learners for halfspaces when the noise is either adversarial or Massart. In independent and concurrent work by (DKK+23), an efficient tester-learner for homogeneous halfspaces achieving error $O(\mathsf{opt}) + \epsilon$ for Gaussian target marginals is also provided, but they do not provide any results for arbitrary strongly log-concave target marginals (see Theorem 5.1) or a guarantee of $\mathsf{opt} + \epsilon$ for Massart noise. In subsequent work by (GKSV23), our techniques were used to provide tester-learners that are not tailored to a single target distribution, but are guaranteed to accept any member of a large family of distributions. Although their main results are more general, their approach crucially extends our

approach here. Moreover, on the technical side, the proof we give here shows how to make use of the moment-matching approach of (GKK23) to provide fully polynomial-time efficient tester-learners, which might be of independent interest.

## 1.1 RELATED WORK

We provide a partial summary of some of the most relevant prior and related work on efficient algorithms for learning halfspaces in the presence of adversarial label or Massart noise, and refer the reader to (BH21) for a survey.

In the distribution-specific agnostic setting where the marginal is assumed to be isotropic and log-concave, (KLS09) showed an algorithm achieving error $O(\mathsf{opt}^{1/3}) + \epsilon$ for the class of origin-centered halfspaces. (ABL17) later obtained $O(\mathsf{opt}) + \epsilon$ using an approach that introduced the principle of iterative *localization*, where the learner focuses attention on a band around a candidate halfspace in order to produce an improved candidate. (Dan15) used this principle to obtain a PTAS for agnostically learning halfspaces under the uniform distribution on the sphere, and (BZ17) extended it to more general $s$-concave distributions. Further works in this line include (YZ17; Zha18; ZSA20; ZL21). (DKTZ20b) introduced the simplest approach yet, based entirely on nonconvex SGD, and showed that it achieves $O(\mathsf{opt}) + \epsilon$ for origin-centered halfspaces over a wide class of structured distributions. Other related works include (DKS18; DKTZ22).

In the Massart noise setting with noise rate bounded by $\eta$, work of (DGT19) gave the first efficient distribution-free algorithm achieving error $\eta + \epsilon$; further improvements and followups include (DKT21; DTK22). However, the optimal error opt achievable by a halfspace may be much smaller than $\eta$, and it has been shown that there are distributions where achieving error competitive with opt as opposed to $\eta$ is computationally hard (DK22; DKMR22). As a result, the distribution-specific setting remains well-motivated for Massart noise. Early distribution-specific algorithms were given by (ABHU15; ABHZ16), but a key breakthrough was the nonconvex SGD approach introduced by (DKTZ20a), which achieved error $\mathsf{opt} + \epsilon$ for origin-centered halfspaces efficiently over a wide range of distributions. This was later generalized by (DKK+22).

## 1.2 TECHNICAL OVERVIEW

Our starting point is the nonconvex optimization approach to learning noisy halfspaces due to (DKTZ20a; DKTZ20b). The algorithms in these works consist of running SGD on a natural nonconvex surrogate $\mathcal{L}_\sigma$ for the 0-1 loss, namely a smooth version of the ramp loss. The key structural property shown is that if the marginal distribution is structured (e.g. log-concave) and the slope of the ramp is picked appropriately, then any $\mathbf{w}$ that has large angle with an optimal $\mathbf{w}^*$ cannot be an approximate stationary point of the surrogate loss $\mathcal{L}_\sigma$, i.e. that $\|\nabla\mathcal{L}_\sigma(\mathbf{w})\|$ must be large. This is proven by carefully analyzing the contributions to the gradient norm from certain critical regions of $\mathrm{span}(\mathbf{w}, \mathbf{w}^*)$, and crucially using the distributional assumption that the probability masses of these regions are proportional to their geometric measures. (See Fig. 3.) In the testable learning setting, the main challenge we face in adapting this approach is checking such a property for the unknown distribution we have access to.

A preliminary observation is that the critical regions of $\mathrm{span}(\mathbf{w}, \mathbf{w}^*)$ that we need to analyze are rectangles, and are hence functions of a small number of halfspaces. Encouragingly, one of the key structural results of the prior work of (GKK23) pertains to "fooling" such functions. Concretely, they show that whenever the true marginal $D_{\mathcal{X}}$ matches moments of degree at most $\widetilde{O}(1/\tau^2)$ with a target $D^*$ that satisfies suitable concentration and anticoncentration properties, then $|\mathbb{E}_{D_{\mathcal{X}}}[f] - \mathbb{E}_{D^*}[f]| \leq \tau$ for any $f$ that is a function of a small number of halfspaces. If we could run such a test and ensure that the probabilities of the critical regions over our empirical marginal are also related to their areas, then we would have a similar stationary point property. However, the difficulty is that since we wish to run in fully polynomial time, we can only hope to fool such functions up to $\tau$ that is a constant. Unfortunately, this is not sufficient to analyze the probability masses of the critical regions we care about as they may be very small.

The chief insight that lets us get around this issue is that each critical region $R$ is in fact of a very specific form, namely a rectangle that is axis-aligned with $\mathbf{w}$: $R = \{\mathbf{x} : \langle\mathbf{w}, \mathbf{x}\rangle \in [-\sigma, \sigma]$ and $\langle\mathbf{v}, \mathbf{x}\rangle \in [\alpha, \beta]\}$ for some values $\alpha, \beta, \sigma$ and some $\mathbf{v}$ orthogonal to $\mathbf{w}$. Moreover, we *know* $\mathbf{w}$, meaning

we can efficiently estimate the probability $\mathbb{P}_{D_{\mathcal{X}}}[\langle \mathbf{w}, \mathbf{x} \rangle \in [-\sigma, \sigma]]$ up to constant multiplicative factors without needing moment tests. Denoting the band $\{\mathbf{x} : \langle \mathbf{w}, \mathbf{x} \rangle \in [-\sigma, \sigma]\}$ by $T$ and writing $\mathbb{P}_{D_{\mathcal{X}}}[R] = \mathbb{P}_{D_{\mathcal{X}}}[\langle \mathbf{v}, \mathbf{x} \rangle \in [\alpha, \beta] \mid \mathbf{x} \in T] \mathbb{P}_{D_{\mathcal{X}}}[T]$, it turns out that we should expect $\mathbb{P}_{D_{\mathcal{X}}}[\langle \mathbf{v}, \mathbf{x} \rangle \in [\alpha, \beta] \mid \mathbf{x} \in T] = \Theta(1)$, as this is what would occur under the structured target distribution $D^*$. (Such a "localization" property is also at the heart of the algorithms for approximately learning halfspaces of, e.g., (ABL17; Dan15).) To check this, it suffices to run tests that ensure that $\mathbb{P}_{D_{\mathcal{X}}}[\langle \mathbf{v}, \mathbf{x} \rangle \in [\alpha, \beta] \mid \mathbf{x} \in T]$ is within an additive constant of this probability under $D^*$.

We can now describe the core of our algorithm (omitting some details such as the selection of the slope of the ramp). First, we run SGD on the surrogate loss $\mathcal{L}$ to arrive at an approximate stationary point and candidate vector $\mathbf{w}$ (technically a list of such candidates). Then, we define the band $T$ based on $\mathbf{w}$, and run tests on the empirical distribution conditioned on $T$. Specifically, we check that the low-degree empirical moments conditioned on $T$ match those of $D^*$ conditioned on $T$, and then apply the structural result of (GKK23) to ensure conditional probabilities of the form $\mathbb{P}_{D_{\mathcal{X}}}[\langle \mathbf{v}, \mathbf{x} \rangle \in [\alpha, \beta] \mid \mathbf{x} \in T]$ match $\mathbb{P}_{D^*}[\langle \mathbf{v}, \mathbf{x} \rangle \in [\alpha, \beta] \mid \mathbf{x} \in T]$ up to a suitable additive constant. This suffices to ensure that even over our empirical marginal, the particular stationary point $\mathbf{w}$ we have is indeed close in angular distance to an optimal $\mathbf{w}^*$.

A final hurdle that remains, often taken for granted under structured distributions, is that closeness in angular distance $\angle(\mathbf{w}, \mathbf{w}^*)$ does not immediately translate to closeness in terms of agreement, $\mathbb{P}[\text{sign}(\langle \mathbf{w}, \mathbf{x} \rangle) \neq \text{sign}(\langle \mathbf{w}^*, \mathbf{x} \rangle)]$, over our unknown marginal. Nevertheless, we show that when the target distribution is Gaussian, we can run polynomial-time tests that ensure that an angle of $\theta = \angle(\mathbf{w}, \mathbf{w}^*)$ translates to disagreement of at most $O(\theta)$. When the target distribution is a general strongly log-concave distribution, we show a slightly weaker relationship: for any $k \in \mathbb{N}$, we can run tests requiring time $d^{\widetilde{O}(k)}$ that ensure that an angle of $\theta$ translates to disagreement of at most $O(\sqrt{k} \cdot \theta^{1-1/k})$. In the Massart noise setting, we can make $\angle(\mathbf{w}, \mathbf{w}^*)$ arbitrarily small, and so obtain our $\text{opt} + \epsilon$ guarantee for any target strongly log-concave distribution in polynomial time. In the adversarial noise setting, we face a more delicate tradeoff and can only make $\angle(\mathbf{w}, \mathbf{w}^*)$ as small as $\Theta(\text{opt})$. When the target distribution is Gaussian, this is enough to obtain final error $O(\text{opt}) + \epsilon$ in polynomial time. When the target distribution is a general strongly log-concave distribution, we instead obtain $\widetilde{O}(\text{opt}) + \epsilon$ in quasipolynomial time.

## 2 PRELIMINARIES

**Notation and setup** Throughout, the domain will be $\mathcal{X} = \mathbb{R}^d$, and labels will lie in $\mathcal{Y} = \{\pm 1\}$. The unknown joint distribution over $\mathcal{X} \times \mathcal{Y}$ that we have access to will be denoted by $D_{\mathcal{X}\mathcal{Y}}$, and its marginal on $\mathcal{X}$ will be denoted by $D_{\mathcal{X}}$. The target marginal on $\mathcal{X}$ will be denoted by $D^*$. We use the following convention for monomials: for a multi-index $\alpha = (\alpha_1, \ldots, \alpha_d) \in \mathbb{Z}_{\geq 0}^d$, $\mathbf{x}^\alpha$ denotes $\prod_i x_i^{\alpha_i}$, and $|\alpha| = \sum_i \alpha_i$ denotes its total degree. We use $\mathcal{C}$ to denote a concept class mapping $\mathbb{R}^d$ to $\{\pm 1\}$, which throughout this paper will be the class of halfspaces or functions of halfspaces over $\mathbb{R}^d$. We use $\text{opt}(\mathcal{C}, D_{\mathcal{X}\mathcal{Y}})$ to denote the optimal error $\inf_{f \in \mathcal{C}} \mathbb{P}_{(\mathbf{x},y) \sim D_{\mathcal{X}\mathcal{Y}}}[f(\mathbf{x}) \neq y]$, or just opt when $\mathcal{C}$ and $D_{\mathcal{X}\mathcal{Y}}$ are clear from context. We recall the definitions of the noise models we consider. In the Massart noise model, the labels satisfy $\mathbb{P}_{y \sim D_{\mathcal{X}\mathcal{Y}}|\mathbf{x}}[y \neq \text{sign}(\langle \mathbf{w}^*, \mathbf{x} \rangle) \mid \mathbf{x}] = \eta(\mathbf{x})$, where $\eta(\mathbf{x}) \leq \eta < \frac{1}{2}$ for all $\mathbf{x}$. In the adversarial label noise or agnostic model, the labels may be completely arbitrary. In both cases, the learner's goal is to produce a hypothesis with error competitive with opt. We now formally define testable learning. The following definition is an equivalent reframing of the original definition (RV23, Def 4), folding the (label-aware) tester and learner into a single tester-learner.

**Definition 2.1** (Testable learning, (RV23)). Let $\mathcal{C}$ be a concept class mapping $\mathbb{R}^d$ to $\{\pm 1\}$. Let $D^*$ be a certain target marginal on $\mathbb{R}^d$. Let $\epsilon, \delta > 0$ be parameters, and let $\psi : [0, 1] \to [0, 1]$ be some function. We say $\mathcal{C}$ can be testably learned w.r.t. $D^*$ up to error $\psi(\text{opt}) + \epsilon$ with failure probability $\delta$ if there exists a tester-learner $A$ meeting the following specification. For any distribution $D_{\mathcal{X}\mathcal{Y}}$ on $\mathbb{R}^d \times \{\pm 1\}$, $A$ takes in a large sample $S$ drawn from $D_{\mathcal{X}\mathcal{Y}}$, and either rejects $S$ or accepts and produces a hypothesis $h : \mathbb{R}^d \to \{\pm 1\}$. Further, the following conditions must be met:

(a) (Soundness.) Whenever $A$ accepts and produces a hypothesis $h$, with probability at least $1 - \delta$ (over the randomness of $S$ and $A$), $h$ must satisfy $\mathbb{P}_{(\mathbf{x},y) \sim D_{\mathcal{X}\mathcal{Y}}}[h(\mathbf{x}) \neq y] \leq \psi(\text{opt}(\mathcal{C}, D_{\mathcal{X}\mathcal{Y}})) + \epsilon$.

(b) (Completeness.) Whenever $D_{\mathcal{X}\mathcal{Y}}$ truly has marginal $D^*$, $A$ must accept with probability at least $1 - \delta$ (over the randomness of $S$ and $A$).

## 3 TESTING PROPERTIES OF STRONGLY LOG-CONCAVE DISTRIBUTIONS

In this section we define the testers that we will need for our algorithm. All the proofs from this section can be found in Appendix B. We begin with a structural lemma that strengthens the key structural result of (GKK23), stated here as Proposition A.3. It states that even when we restrict an isotropic strongly log-concave $D^*$ to a band around the origin, moment matching suffices to fool functions of halfspaces whose weights are orthogonal to the normal of the band.

**Proposition 3.1.** *Let $D^*$ be an isotropic strongly log-concave distribution. Let $\mathbf{w} \in \mathbb{S}^{d-1}$ be any fixed direction. Let $p$ be a constant. Let $f : \mathbb{R}^d \to \mathbb{R}$ be a function of $p$ halfspaces of the form in Eq. (A.2), with the additional restriction that its weights $\mathbf{v}^i \in \mathbb{S}^{d-1}$ satisfy $\langle \mathbf{v}^i, \mathbf{w} \rangle = 0$ for all $i$. For some $\sigma \in [0, 1]$, let $T$ denote the band $\{\mathbf{x} : |\langle \mathbf{w}, \mathbf{x} \rangle| \leq \sigma\}$. Let $D$ be any distribution such that $D_{|T}$ matches moments of degree at most $k = \widetilde{O}(1/\tau^2)$ with $D^*_{|T}$ up to an additive slack of $d^{-\widetilde{O}(k)}$. Then $|\mathbb{E}_{D^*}[f \mid T] - \mathbb{E}_D[f \mid T]| \leq \tau$.*

We now describe some of the testers that we use. First, we need a tester that ensures that the distribution is concentrated in every single direction. More formally, the tester checks that the moments of the distribution along any direction are small.

**Proposition 3.2.** *For any isotropic strongly log-concave $D^*$, there exists some constants $C_1$ and a tester $T_1$ that takes a set $S \subseteq \mathbb{R}^d \times \{\pm 1\}$, an even $k \in \mathbb{N}$, a parameter $\delta \in (0, 1)$ and runs and in time $\operatorname{poly}\left(d^k, |S|, \log \frac{1}{\delta}\right)$. Let $D$ denote the uniform distribution over $S$. If $T_1$ accepts, then for any $\mathbf{v} \in \mathbb{S}^{d-1}$*

$$\mathbb{E}_{(\mathbf{x},y)\sim D}[(\langle \mathbf{v}, \mathbf{x} \rangle)^k] \leq (C_1 k)^{k/2}. \tag{3.1}$$

*Moreover, if $S$ is obtained by taking at least $\left(d^k, \left(\log \frac{1}{\delta}\right)^k\right)^{C_1}$ i.i.d. samples from a distribution whose $\mathbb{R}^d$-marginal is $D^*$, the test $T_1$ passes with probability at least $1 - \delta$.*

Secondly, we will use a tester that makes sure the distribution is not concentrated too close to a specific hyperplane. This is one of the properties we will need to use in order to employ the localization technique of (ABL17).

**Proposition 3.3.** *For any isotropic strongly log-concave $D^*$, there exist some constants $C_2, C_3$ and a tester $T_2$ that takes a set $S \subseteq \mathbb{R}^d \times \{\pm 1\}$ a vector $\mathbf{w} \in \mathbb{S}^{d-1}$, parameters $\sigma, \delta \in (0, 1)$ and runs in time $\operatorname{poly}\left(d, |S|, \log \frac{1}{\delta}\right)$. Let $D$ denote the uniform distribution over $S$. If $T_2$ accepts, then*

$$\mathbb{P}_{(\mathbf{x},y)\sim D}[|\langle \mathbf{w}, \mathbf{x} \rangle| \leq \sigma] \in (C_2\sigma, C_3\sigma). \tag{3.2}$$

*Moreover, if $S$ is obtained by taking at least $\frac{100}{K_1\sigma^2} \log\left(\frac{1}{\delta}\right)$ i.i.d. samples from a distribution whose $\mathbb{R}^d$-marginal is $D^*$, the test $T_2$ passes with probability at least $1 - \delta$.*

Finally, in order to use the localization idea of (ABL17) in a manner similar to (DKTZ20b), we need to make sure that the distribution is well-behaved also within a band around to a certain hyperplane. The main property of the distribution that we establish is that functions of constantly many halfspaces have expectations very close to what they would be under our distributional assumption. As we show later in this work, having the aforementioned property allows us to derive many other properties that strongly log-concave distributions have, including many of the key properties that make the localization technique successful.

**Proposition 3.4.** *For any isotropic strongly log-concave $D^*$ and a constant $C_4$, there exists a constant $C_5$ and a tester $T_3$ that takes a set $S \subseteq \mathbb{R}^d \times \{\pm 1\}$ a vector $\mathbf{w} \in \mathbb{S}^{d-1}$, parameters $\sigma, \tau \delta \in (0, 1)$ and runs in time $\operatorname{poly}\left(d^{\tilde{O}\left(\frac{1}{\tau^2}\right)}, \frac{1}{\sigma}, |S|, \log \frac{1}{\delta}\right)$. Let $D$ denote the uniform distribution over $S$, let $T$ denote the band $\{\mathbf{x} : |\langle \mathbf{w}, \mathbf{x} \rangle| \leq \sigma\}$ and let $\mathcal{F}_{\mathbf{w}}$ denote the set $\{\pm 1\}$-valued functions of $C_4$ halfspaces whose weight vectors are orthogonal to $\mathbf{w}$. If $T_3$ accepts, then*

$$\max_{f \in \mathcal{F}_{\mathbf{w}}} \left| \mathbb{E}_{\mathbf{x}\sim D^*}[f(\mathbf{x}) \mid \mathbf{x} \in T] - \mathbb{E}_{(\mathbf{x},y)\sim D}[f(\mathbf{x}) \mid \mathbf{x} \in T] \right| \leq \tau, \tag{3.3}$$

$$\max_{\mathbf{v}\in\mathbb{S}^{d-1}:\ \langle\mathbf{v},\mathbf{w}\rangle=0}\left|\underset{\mathbf{x}\sim D^*}{\mathbb{E}}\left[(\langle\mathbf{v},\mathbf{x}\rangle)^2\mid\mathbf{x}\in T\right]-\underset{(\mathbf{x},y)\sim D}{\mathbb{E}}\left[(\langle\mathbf{v},\mathbf{x}\rangle)^2\mid\mathbf{x}\in T\right]\right|\le\tau. \qquad (3.4)$$

*Moreover, if $S$ is obtained by taking at least $\left(\frac{1}{\tau}\cdot\frac{1}{\sigma}\cdot d^{\frac{1}{\tau^2}\log^{C_5}\left(\frac{1}{\tau}\right)}\cdot\left(\log\frac{1}{\delta}\right)^{\frac{1}{\tau^2}\log^{C_5}\left(\frac{1}{\tau}\right)}\right)^{C_5}$ i.i.d. samples from a distribution whose $\mathbb{R}^d$-marginal is $D^*$, the test $T_3$ passes w.p. at least $1-\delta$.*

## 4 Testably learning halfspaces with Massart noise

In this section we prove that we can testably learn halfspaces with Massart noise with respect to isotropic strongly log-concave distributions (see Definition A.1).

**Theorem 4.1** (Tester-Learner for Halfspaces with Massart Noise). *Let $D_{\mathcal{X}\mathcal{Y}}$ be a distribution over $\mathbb{R}^d\times\{\pm1\}$ and let $D^*$ be an isotropic strongly log-concave distribution over $\mathbb{R}^d$. Let $\mathcal{C}$ be the class of origin centered halfspaces in $\mathbb{R}^d$. Then, for any $\eta<1/2$, $\epsilon>0$ and $\delta\in(0,1)$, there exists an algorithm (Algorithm 1) that testably learns $\mathcal{C}$ w.r.t. $D^*$ up to excess error $\epsilon$ and error probability at most $\delta$ in the Massart noise model with rate at most $\eta$, using time and a number of samples from $D_{\mathcal{X}\mathcal{Y}}$ that are polynomial in $d,1/\epsilon,\frac{1}{1-2\eta}$ and $\log(1/\delta)$.*

---

**Algorithm 1:** Tester-learner for halfspaces

**Input:** Training sets $S_1,S_2$, parameters $\sigma,\delta,\alpha$
**Output:** A near-optimal weight vector $\mathbf{w}$, or rejection
Run PSGD on the empirical loss $\mathcal{L}_\sigma$ over $S_1$ to get a list $L$ of candidate vectors.
Test whether $L$ contains an $\alpha$-approximate stationary point $\mathbf{w}$ of the empirical loss $\mathcal{L}_\sigma$ over $S_2$.
  Reject if no such $\mathbf{w}$ exists.
**for** *each candidate $\mathbf{w}'$ in $\{\mathbf{w},-\mathbf{w}\}$* **do**
    Let $B_{\mathbf{w}'}(\sigma)$ denote the band $\{\mathbf{x}:|\langle\mathbf{w}',\mathbf{x}\rangle|\le\sigma\}$. Let $\mathcal{F}_{\mathbf{w}'}$ denote the class of functions of
      at most two halfspaces with weights orthogonal to $\mathbf{w}'$.
    Let $\delta'=\Theta(\delta)$.
    Run $T_1(S_2,k=2,\delta)$ to verify that the empirical marginal is approximately isotropic.
      Reject if $T_1$ rejects.
    Run $T_2(S_2,\mathbf{w}',\sigma,\delta')$ to verify that $\mathbb{P}_S[B_{\mathbf{w}'}(\sigma)]=\Theta(\sigma)$. Reject if $T_2$ rejects.
    Run $T_3(S_2,\mathbf{w}',\sigma=\sigma/6,\tau,\delta')$ and $T_3(S,\mathbf{w}',\sigma=\sigma/2,\tau,\delta')$ for a suitable constant $\tau$ to
      verify that the empirical distribution conditioned on $B_{\mathbf{w}'}(\sigma/6)$ and $B_{\mathbf{w}'}(\sigma/2)$ fools $\mathcal{F}_{\mathbf{w}'}$
      up to $\tau$. Reject if $T_3$ rejects.
    Estimate the empirical error of $\mathbf{w}'$ on $S$.
If all tests have accepted, output $\mathbf{w}'\in\{\mathbf{w},-\mathbf{w}\}$ with the best empirical error.

---

To show our result, we revisit the approach of (DKTZ20a) for learning halfspaces with Massart noise under well-behaved distributions. Their result is based on the idea of minimizing a surrogate loss that is non convex, but whose stationary points correspond to halfspaces with low error. They also require that their surrogate loss is sufficiently smooth, so that one can find a stationary point efficiently. While the distributional assumptions that are used to demonstrate that stationary points of the surrogate loss can be discovered efficiently are mild, the main technical lemma, which demostrates that any stationary point suffices, requires assumptions that are not necessarily testable. We establish a label-dependent approach for testing, making use of tests that are applied during the course of our algorithm.

We consider a slightly different surrogate loss than the one used in (DKTZ20a). In particular, for $\sigma>0$, we let

$$\mathcal{L}_\sigma(\mathbf{w})=\underset{(\mathbf{x},y)\sim D_{\mathcal{X}\mathcal{Y}}}{\mathbb{E}}\left[\ell_\sigma\left(-y\frac{\langle\mathbf{w},\mathbf{x}\rangle}{\|\mathbf{w}\|_2}\right)\right], \qquad (4.1)$$

where $\ell_\sigma:\mathbb{R}\to[0,1]$ is a smooth approximation to the ramp function with the properties described in Proposition C.1 (see Appendix C), obtained using a piecewise polynomial of degree 3. Unlike the standard logistic function, our loss function has derivative exactly 0 away from the origin (for $|t|>\sigma/2$). This makes the analysis of the gradient of $\mathcal{L}_\sigma$ easier, since the contribution from points lying outside a certain band is exactly 0.

The smoothness allows us to run PSGD to obtain stationary points efficiently, and we now state the convergence lemma we need.

**Proposition 4.2** (PSGD Convergence, Lemmas 4.2 and B.2 in (DKTZ20a)). *Let $\mathcal{L}_\sigma$ be as in Equation equation 4.1 with $\sigma \in (0, 1]$, $\ell_\sigma$ as described in Proposition C.1 and $D_{\mathcal{X}\mathcal{Y}}$ such that the marginal $D_{\mathcal{X}}$ on $\mathbb{R}^d$ satisfies Property equation 3.1 for $k = 2$. Then, for any $\epsilon > 0$ and $\delta \in (0, 1)$, there is an algorithm whose time and sample complexity is $O(\frac{d}{\sigma^4} + \frac{\log(1/\delta)}{\epsilon^4 \sigma^4})$, which, having access to samples from $D_{\mathcal{X}\mathcal{Y}}$, outputs a list $L$ of vectors $\mathbf{w} \in \mathbb{S}^{d-1}$ with $|L| = O(\frac{d}{\sigma^4} + \frac{\log(1/\delta)}{\epsilon^4 \sigma^4})$ so that there exists $\mathbf{w} \in L$ with*

$$\|\nabla_{\mathbf{w}} \mathcal{L}_\sigma(\mathbf{w})\|_2 \leq \epsilon, \text{ with probability at least } 1 - \delta.$$

*In particular, the algorithm performs Stochastic Gradient Descent on $\mathcal{L}_\sigma$ Projected on $\mathbb{S}^{d-1}$ (PSGD).*

It now suffices to show that, upon performing PSGD on $\mathcal{L}_\sigma$, for some appropriate choice of $\sigma$, we acquire a list of vectors that testably contain a vector which is approximately optimal. We first prove the following lemma, whose distributional assumptions are relaxed compared to the corresponding structural Lemma 3.2 of (DKTZ20a). In particular, instead of requiring the marginal distribution to be "well-behaved", we assume that the quantities of interest (for the purposes of our proof) have expected values under the true marginal distribution that are close, up to multiplicative factors, to their expected values under some "well-behaved" (in fact, strongly log-concave) distribution. While some of the quantities of interest have values that are miniscule and estimating them up to multiplicative factors could be too costly, it turns out that the source of their vanishing scaling can be completely attributed to factors of the form $\mathbb{P}[|\langle \mathbf{w}, \mathbf{x} \rangle| \leq \sigma]$ (where $\sigma$ is small), which, due to standard concentration arguments, can be approximated up to multiplicative factors, given $\mathbf{w} \in \mathbb{S}^{d-1}$ and $\sigma > 0$ (see Proposition 3.3). As a result, we may estimate the remaining factors up to sufficiently small additive constants (see Proposition 3.4) to get multiplicative overall closeness to the "well behaved" baseline. We defer the proof of the following Lemma to Appendix C.1.

**Lemma 4.3.** *Let $\mathcal{L}_\sigma$ be as in Equation equation 4.1 with $\sigma \in (0, 1]$, $\ell_\sigma$ as described in Proposition C.1, let $\mathbf{w} \in \mathbb{S}^{d-1}$ and consider $D_{\mathcal{X}\mathcal{Y}}$ such that the marginal $D_{\mathcal{X}}$ on $\mathbb{R}^d$ satisfies Properties equation 3.2 and equation 3.3 for $C_4 = 2$ and accuracy $\tau$. Let $\mathbf{w}^* \in \mathbb{S}^{d-1}$ define an optimum halfspace and let $\eta < 1/2$ be an upper bound on the rate of the Massart noise. Then, there are constants $c_1, c_2, c_3 > 0$ such that if $\|\nabla_{\mathbf{w}} \mathcal{L}_\sigma(\mathbf{w})\|_2 < c_1(1 - 2\eta)$ and $\tau \leq c_2$, then*

$$\angle(\mathbf{w}, \mathbf{w}^*) \leq \frac{c_3}{1 - 2\eta} \cdot \sigma \quad or \quad \angle(-\mathbf{w}, \mathbf{w}^*) \leq \frac{c_3}{1 - 2\eta} \cdot \sigma$$

Combining Proposition 4.2 and Lemma 4.3, we get that for any choice of the parameter $\sigma \in (0, 1]$, by running PSGD on $\mathcal{L}_\sigma$, we can construct a list of vectors of polynomial size (in all relevant parameters) that testably contains a vector that is close to the optimum weight vector. In order to link the zero-one loss to the angular similarity between a weight vector and the optimum vector, we use the following Proposition (for the proof, see Appendix C.2).

**Proposition 4.4.** *Let $D_{\mathcal{X}\mathcal{Y}}$ be a distribution over $\mathbb{R}^d \times \{\pm 1\}$, $\mathbf{w}^* \in \arg\min_{\mathbf{w} \in \mathbb{S}^{d-1}} \mathbb{P}_{D_{\mathcal{X}\mathcal{Y}}}[y \neq \text{sign}(\langle \mathbf{w}, \mathbf{x} \rangle)]$ and $\mathbf{w} \in \mathbb{S}^{d-1}$. Then, for any $\theta \geq \angle(\mathbf{w}, \mathbf{w}^*)$, $\theta \in [0, \pi/4]$, if the marginal $D_{\mathcal{X}}$ on $\mathbb{R}^d$ satisfies Property equation 3.1 for $C_1 > 0$ and some even $k \in \mathbb{N}$ and Property equation 3.2 with $\sigma$ set to $(C_1 k)^{\frac{k}{2(k+1)}} \cdot (\tan\theta)^{\frac{k}{k+1}}$, then, there exists a constant $c > 0$ such that the following is true.*

$$\mathbb{P}_{D_{\mathcal{X}\mathcal{Y}}}[y \neq \text{sign}(\langle \mathbf{w}, \mathbf{x} \rangle)] \leq \text{opt} + c \cdot k^{1/2} \cdot \theta^{1 - \frac{1}{k+1}}.$$

*Proof of Theorem 4.1.* Throughout the proof we consider $\delta'$ to be a sufficiently small polynomial in all the relevant parameters. Each of the failure events will have probability at least $\delta'$ and their number will be polynomial in all the relevant parameters, so by the union bound, we may pick $\delta'$ so that the probability of failure is at most $\delta$.

The algorithm we run is Algorithm 1, with appropriate selection of parameters and given samples $S_1$, $S_2$, each of which are sufficiently large sets of independent samples from the true unknown distribution $D_{\mathcal{X}\mathcal{Y}}$. For some $\sigma \in (0, 1]$ to be defined later, we run PSGD on the empirical loss $\mathcal{L}_\sigma$ over $S_1$ as described in Proposition 4.2 with $\epsilon = c_1(1 - 2\eta)\sigma/4$, where $c_1$ is given by Lemma 4.3.

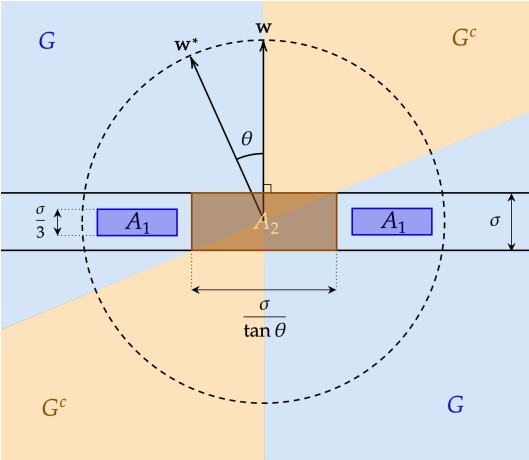

Figure 1: Critical regions in the proofs of main structural lemmas (Lemmas 4.3, 5.2). We analyze the contributions of the regions labeled $A_1, A_2$ to the quantities $A_1, A_2$ in the proofs. Specifically, the regions $A_1$ (which have height $\sigma/3$ so that the value of $\ell'_\sigma(\mathbf{x_w})$ for any $\mathbf{x}$ in these regions is exactly $1/\sigma$, by Proposition C.1) form a subset of the region $\mathcal{G}$, and their probability mass under $D_\mathcal{X}$ is (up to a multiplicative factor) a lower bound on the quantity $A_1$ (see Eq equation C.3). Similarly, the region $A_2$ is a subset of the intersection of $\mathcal{G}^c$ with the band of height $\sigma$, and has probability mass that is (up to a multiplicative factor) an upper bound on the quantity $A_2$ (see Eq equation C.4).

By Proposition 4.2, we get a list $L$ of vectors $\mathbf{w} \in \mathbb{S}^{d-1}$ with $|L| = \text{poly}(d, 1/\sigma)$ such that there is $\mathbf{w} \in L$ with $\|\nabla_\mathbf{w}\mathcal{L}_\sigma(\mathbf{w})\|_2 < \frac{1}{2}c_1(1 - 2\eta)$ under the true distribution, if the marginal is isotropic.

Having acquired the list $L$ using sample $S_1$, we use the independent samples in $S_2$ to test whether $L$ contains an approximately stationary point of the empirical loss on $S_2$. If this is not the case, then we may safely reject: for large enough $|S_1|$, if the distribution is indeed isotropic strongly logconcave, there is an approximate stationary of the population loss in $L$ and if $|S_2|$ is large enough, the gradient of the empirical loss on $S_2$ will be close to the gradient of the population loss on each of the elements of $L$, due to appropriate concentration bounds for log-concave distributions as well as the fact that the elements of $L$ are independent from $S_2$. For the following, let $\mathbf{w}$ be a point such that $\|\nabla_\mathbf{w}\mathcal{L}_\sigma(\mathbf{w})\|_2 < c_1(1 - 2\eta)$ under the empirical distribution over $S_2$

In Lemma 4.3 and Proposition 4.4 we have identified certain properties of the marginal distribution that are sufficient for our purposes, given that $L$ contains an approximately stationary point of the empirical (surrogate) loss on $S_2$. Our testers $T_1, T_2, T_3$ verify that these properties hold for the empirical marginal over our sample $S_2$, and it will be convenient to analyze the optimality of our algorithm purely over $S_2$. In particular, we will need to require that $|S_2|$ is sufficiently large, so that when the true marginal is indeed the target $D^*$, our testers succeed with high probability (for the corresponding sample complexity, see Propositions 3.2, 3.3 and 3.4). Moreover, by standard generalization theory, since the VC dimension of halfspaces is only $O(d)$ and for us $|S_2|$ is a large $\text{poly}(d, 1/\epsilon)$, both the error of our final output and the optimal error over $S_2$ will be close to that over $D_{\mathcal{XY}}$. So in what follows, we will abuse notation and refer to the uniform distribution over $S_2$ as $D_{\mathcal{XY}}$ and the optimal error over $S_2$ simply as opt.

We proceed with some basic tests. Throughout the rest of the algorithm, whenever a tester fails, we reject, otherwise we proceed. First, we run testers $T_2$ with inputs $(\mathbf{w}, \sigma/2, \delta')$ and $(\mathbf{w}, \sigma/6, \delta')$ (Proposition 3.3) and $T_3$ with inputs $(\mathbf{w}, \sigma/2, c_2, \delta')$ and with $(\mathbf{w}, \sigma/6, c_2, \delta')$ (Proposition 3.4, $c_2$ as defined in Lemma 4.3). This ensures that for the approximate stationary point $\mathbf{w}$ of the $\mathcal{L}_\sigma$, the probability within the band $B_\mathbf{w}(\sigma/2) = \{\mathbf{x} : |\langle \mathbf{w}, \mathbf{x} \rangle| \le \sigma/2\}$ is $\Theta(\sigma)$ (and similarly for $B_\mathbf{w}(\sigma/6)$) and moreover that our marginal conditioned on each of the bands fools (up to an additive constant) functions of halfspaces with weights orthogonal to $\mathbf{w}$. As a result, we may apply Lemma 4.3 to $\mathbf{w}$ and form a list of 2 vectors $\{\mathbf{w}, -\mathbf{w}\}$ which contains some $\mathbf{w}'$ with $\angle(\mathbf{w}', \mathbf{w}^*) \le c_2\sigma/(1 - 2\eta)$ (where $c_3$ is as defined in Lemma 4.3).

We run $T_1$ (Proposition 3.2) with $k = 2$ to verify that the marginals are approximately isotropic and we use $T_2$ once again, with appropriate parameters for each $\mathbf{w}$ and its negation, to apply Proposition 4.4 and get that $\{\mathbf{w}, -\mathbf{w}\}$ contains a vector $\mathbf{w}'$ with

$$\mathbb{P}_{D_{\mathcal{X}\mathcal{Y}}}[y \neq \text{sign}(\langle \mathbf{w}', \mathbf{x} \rangle)] \leq \text{opt} + c \cdot \theta^{2/3}, \text{ where } \measuredangle(\mathbf{w}', \mathbf{w}^*) \leq \theta := c_2 \sigma / \sqrt{1 - 2\eta}$$

By picking $\sigma = \Theta(\epsilon^{3/2}(1 - 2\eta))$ we have $\mathbb{P}_{D_{\mathcal{X}\mathcal{Y}}}[y \neq \text{sign}(\langle \mathbf{w}', \mathbf{x} \rangle)] \leq \text{opt} + \epsilon$.

However, we do not know which of the weight vectors in $\{\mathbf{w}, -\mathbf{w}\}$ is the one guaranteed to achieve small error. In order to discover this vector, we estimate the probability of error of each of the corresponding halfspaces (which can be done efficiently, due to Hoeffding's bound) and pick the one with the smallest error. This final step does not require any distributional assumptions and we do not need to perform any further tests. $\qquad \square$

## 5 TESTABLY LEARNING HALFSPACES IN THE AGNOSTIC SETTING

In this section, we provide our result on efficiently and testably learning halfspaces in the agnostic setting with respect to isotropic strongly log-concave target marginals. We defer the proofs to Appendix D. The algorithm we use is once more Algorithm 1, but we call it multiple times for different choices of the parameter $\sigma$, reject if any call rejects and output the vector that achieved the minimum empirical error overall, otherwise. Also, the tester $T_1$ is called for a general $k$ (not necessarily $k = 2$).

**Theorem 5.1** (Efficient Tester-Learner for Halfspaces in the Agnostic Setting)**.** *Let $D_{\mathcal{X}\mathcal{Y}}$ be a distribution over $\mathbb{R}^d \times \{\pm 1\}$ and let $D^*$ be a strongly log-concave distribution over $\mathbb{R}^d$ (Definition A.1). Let $\mathcal{C}$ be the class of origin centered halfspaces in $\mathbb{R}^d$. Then, for any even $k \in \mathbb{N}$, any $\epsilon > 0$ and $\delta \in (0, 1)$, there exists an algorithm that agnostically testably learns $\mathcal{C}$ w.r.t. $D^*$ up to error $O(k^{1/2} \cdot \text{opt}^{1 - \frac{1}{k+1}}) + \epsilon$, where $\text{opt} = \min_{\mathbf{w} \in \mathbb{S}^{d-1}} \mathbb{P}_{D_{\mathcal{X}\mathcal{Y}}}[y \neq \text{sign}(\langle \mathbf{w}, \mathbf{x} \rangle)]$, and error probability at most $\delta$, using time and a number of samples from $D_{\mathcal{X}\mathcal{Y}}$ that are polynomial in $d^{\tilde{O}(k)}, (1/\epsilon)^{\tilde{O}(k)}$ and $(\log(1/\delta))^{O(k)}$. In particular, by picking some appropriate $k \leq \log^2 d$, we obtain error $\tilde{O}(\text{opt}) + \epsilon$ in quasipolynomial time and sample complexity, i.e. $\text{poly}(2^{\text{polylog} d}, (\frac{1}{\epsilon})^{\text{polylog} d})$.*

To prove Theorem 5.1, we may follow a similar approach as the one we used for the case of Massart noise. However, in this case, the main structural lemma regarding the quality of the stationary points involves an additional requirement about the parameter $\sigma$. In particular, $\sigma$ cannot be arbitrarily small with respect to the error of the optimum halfspace, because, in this case, there is no upper bound on the amount of noise that any specific point $\mathbf{x}$ might be associated with. As a result, picking $\sigma$ to be arbitrarily small would imply that our algorithm only considers points that lie within a region that has arbitrarily small probability and can hence be completely corrupted with the adversarial opt budget. On the other hand, the polynomial slackness that the testability requirement introduces (through Proposition 4.4) between the error we achieve and the angular distance guarantee we can get via finding a stationary point of $\mathcal{L}_\sigma$ (which is now coupled with opt), appears to the exponent of the guarantee we achieve in Theorem 5.1.

**Lemma 5.2.** *Let $\mathcal{L}_\sigma$ be as in Equation equation 4.1 with $\sigma \in (0, 1]$, $\ell_\sigma$ as described in Proposition C.1, let $\mathbf{w} \in \mathbb{S}^{d-1}$ and consider $D_{\mathcal{X}\mathcal{Y}}$ such that the marginal $D_{\mathcal{X}}$ on $\mathbb{R}^d$ satisfies Properties equation 3.2, equation 3.3 and equation 3.4 for $\mathbf{w}$ with $C_4 = 2$ and accuracy parameter $\tau$. Let opt be the minimum error achieved by some origin centered halfspace and let $\mathbf{w}^* \in \mathbb{S}^{d-1}$ be a corresponding vector. Then, there are constants $c_1, c_2, c_3, c_4 > 0$ such that if $\text{opt} \leq c_1\sigma$, $\|\nabla_{\mathbf{w}} \mathcal{L}_\sigma(\mathbf{w})\|_2 < c_2$, and $\tau \leq c_3$ then either $\measuredangle(\mathbf{w}, \mathbf{w}^*) \leq c_4\sigma$ or $\measuredangle(-\mathbf{w}, \mathbf{w}^*) \leq c_4\sigma$.*

We obtain our main result for Gaussian target marginals by refining Proposition 4.4 for the specific case when the target marginal distribution $D^*$ is the standard multivariate Gaussian distribution. The algorithm for the Gaussian case is similar to the one of Theorem 5.1, but it runs different tests for the improved version (see Proposition D.1) of Proposition 4.4.

**Theorem 5.3.** *In Theorem 5.1, if $D^*$ is the standard Gaussian in $d$ dimensions, we obtain error $O(\text{opt}) + \epsilon$ in polynomial time and sample complexity, i.e. $\text{poly}(d, 1/\epsilon, \log(1/\delta))$.*

ACKNOWLEDGMENTS

We wish to thank the anonymous reviewers of ICLR 2024 for their constructive feedback. Aravind Gollakota was at UT Austin while this work was done, supported by NSF award AF-1909204 and the NSF AI Institute for Foundations of Machine Learning (IFML). Adam R. Klivans was supported by NSF award AF-1909204 and the NSF AI Institute for Foundations of Machine Learning (IFML). Konstantinos Stavropoulos was supported by NSF award AF-1909204, the NSF AI Institute for Foundations of Machine Learning (IFML), and by scholarships from Bodossaki Foundation and Leventis Foundation. Arsen Vasilyan was supported in part by NSF awards CCF-2006664, DMS-2022448, CCF-1565235, CCF-1955217, CCF-2310818, Big George Fellowship and Fintech@CSAIL. Part of this work was done while Arsen Vasilyan was visiting UT Austin.

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

## A  STRONGLY LOG-CONCAVE DISTRIBUTIONS

We also formally define the class of *strongly log-concave* distributions, which is the class that our target marginal $D^*$ is allowed to belong to, and collect some useful properties of such distributions. We will state the definition for isotropic $D^*$ (i.e. with mean 0 and covariance $I$) for simplicity.

**Definition A.1** (Strongly log-concave distribution, see e.g. (SW14, Def 2.8)). *We say an isotropic distribution $D^*$ on $\mathbb{R}^d$ is strongly log-concave if the logarithm of its density $q$ is a strongly concave function. Equivalently, $q$ can be written as*

$$q(\mathbf{x}) = r(\mathbf{x})\gamma_{\kappa^2 I}(\mathbf{x}) \tag{A.1}$$

*for some log-concave function $r$ and some constant $\kappa > 0$, where $\gamma_{\kappa^2 I}$ denotes the density of the spherical Gaussian $\mathcal{N}(0, \kappa^2 I)$.*

**Proposition A.2** (see e.g. (SW14)). *Let $D^*$ be an isotropic strongly log-concave distribution on $\mathbb{R}^d$ with density $q$.*

(a) *Any orthogonal projection of $D^*$ onto a subspace is also strongly log-concave.*

(b) *There exist constants $U, R$ such that $q(\mathbf{x}) \leq U$ for all $\mathbf{x}$, and $q(x) \geq 1/U$ for all $\|\mathbf{x}\| \leq R$.*

(c) *There exist constants $U'$ and $\kappa$ such that $q(\mathbf{x}) \leq U'\gamma_{\kappa^2 I}(\mathbf{x})$ for all $\mathbf{x}$.*

(d) *There exist constants $K_1, K_2$ such that for any $\sigma \in [0, 1]$ and any $\mathbf{v} \in \mathbb{S}^{d-1}$, $\mathbb{P}[|\langle \mathbf{v}, \mathbf{x}\rangle| \leq \sigma] \in (K_1\sigma, K_2\sigma)$.*

(e) *There exists a constant $K_3$ such that for any $k \in \mathbb{N}$, $\mathbb{E}[|\langle \mathbf{v}, \mathbf{x}\rangle|^k] \leq (K_3 k)^{k/2}$.*

(f) *Let $\alpha = (\alpha_1, \ldots, \alpha_d) \in \mathbb{Z}_{\geq 0}^d$ be a multi-index with total degree $|\alpha| = \sum_i \alpha_i = k$, and let $\mathbf{x}^\alpha = \prod_i x_i^{\alpha_i}$. There exists a constant $K_4$ such that for any such $\alpha$, $\mathbb{E}[|\mathbf{x}^\alpha|] \leq (K_4 k)^{k/2}$.*

For (a), see e.g. (SW14, Thm 3.7). The other properties follow readily from Eq. (A.1), which allows us to treat the density as subgaussian.

A key structural fact that we will need about strongly log-concave distributions is that approximately matching moments of degree at most $\widetilde{O}(1/\tau^2)$ with such a $D^*$ is sufficient to fool any function of a constant number of halfspaces up to an additive $\tau$.

**Proposition A.3** (Variant of (GKK23, Thm 5.6)). *Let $p$ be a fixed constant, and let $\mathcal{F}$ be the class of all functions of $p$ halfspaces mapping $\mathbb{R}^d$ to $\{\pm 1\}$ of the form*

$$f(\mathbf{x}) = g\left(\text{sign}(\langle \mathbf{v}^1, \mathbf{x}\rangle + \theta_1), \ldots, \text{sign}(\langle \mathbf{v}^p, \mathbf{x}\rangle + \theta_p)\right) \tag{A.2}$$

*for some $g : \{\pm 1\}^p \to \{\pm 1\}$ and weights $\mathbf{v}^i \in \mathbb{S}^{d-1}$. Let $D^*$ be any target marginal such that for every $i$, the projection $\langle \mathbf{v}^i, \mathbf{x}\rangle$ has subgaussian tails and is anticoncentrated: (a) $\mathbb{P}[|\langle \mathbf{v}^i, \mathbf{x}\rangle| > t] \leq \exp(-\Theta(t^2))$, and (b) for any interval $[a, b]$, $\mathbb{P}[\langle \mathbf{v}^i, \mathbf{x}\rangle \in [a, b]] \leq \Theta(|b - a|)$. Let $D$ be any distribution such that for all monomials $\mathbf{x}^\alpha = \prod_i x_i^{\alpha_i}$ of total degree $|\alpha| = \sum_i \alpha_i \leq k$,*

$$\left|\mathbb{E}_{D^*}[\mathbf{x}^\alpha] - \mathbb{E}_{D}[\mathbf{x}^\alpha]\right| \leq \left(\frac{c|\alpha|}{d\sqrt{k}}\right)^{|\alpha|}$$

*for some sufficiently small constant $c$ (in particular, it suffices to have $d^{-\widetilde{O}(k)}$ moment closeness for every $\alpha$). Then*

$$\max_{f \in \mathcal{F}} \left|\mathbb{E}_{D^*}[f] - \mathbb{E}_{D}[f]\right| \leq \widetilde{O}\left(\frac{1}{\sqrt{k}}\right).$$

Note that this is a variant of the original statement of (GKK23, Thm 5.6), which requires that the 1D projection of $D^*$ along *any* direction satisfy suitable concentration and anticoncentration. Indeed, an inspection of their proof reveals that it suffices to verify these properties for projections only along the directions $\{\mathbf{v}^i\}_{i \in [p]}$ as opposed to all directions. This is because to fool a function $f$ of the form above, their proof only analyzes the projected distribution $(\langle \mathbf{v}^1, \mathbf{x}\rangle, \ldots, \langle \mathbf{v}^p, \mathbf{x}\rangle)$ on $\mathbb{R}^p$, and requires only concentration and anticoncentration for each individual projection $\langle \mathbf{v}^i, \mathbf{x}\rangle$.

# B PROOFS FOR SECTION 3

## B.1 PROOF OF PROPOSITION 3.1

Our plan is to apply Proposition A.3. To do so, we must verify that $D^*_{|T}$ satisfies the assumptions required. In particular, it suffices to verify that the 1D projection along any direction orthogonal to $\mathbf{w}$ has subgaussian tails and is anticoncentrated. Let $\mathbf{v} \in \mathbb{S}^{d-1}$ be any direction that is orthogonal to $\mathbf{w}$. By Proposition A.2(d), we may assume that $\mathbb{P}_{D^*}[T] \geq \Omega(\sigma)$.

To verify subgaussian tails, we must show that for any $t$, $\mathbb{P}_{D^*_{|T}}[|\langle \mathbf{v}, \mathbf{x} \rangle| > t] \leq \exp(-Ct^2)$ for some constant $C$. The main fact we use is Proposition A.2(c), i.e. that any strongly log-concave density is pointwise upper bounded by a Gaussian density times a constant. Write

$$\mathbb{P}_{D^*_{|T}}[|\langle \mathbf{v}, \mathbf{x} \rangle| > t] = \frac{\mathbb{P}_{D^*}[\langle \mathbf{v}, \mathbf{x} \rangle > t \text{ and } \langle \mathbf{w}, \mathbf{x} \rangle \in [-\sigma, \sigma]]}{\mathbb{P}_{D^*}[\langle \mathbf{w}, \mathbf{x} \rangle \in [-\sigma, \sigma]]}.$$

The claim now follows from the fact that the numerator is upper bounded by a constant times the corresponding probability under a Gaussian density, which is at most $O(\exp(-C't^2)\sigma)$ for some constant $C'$, and that the denominator is $\Omega(\sigma)$.

To check anticoncentration, for any interval $[a, b]$, write

$$\mathbb{P}_{D^*_{|T}}[\langle \mathbf{v}, \mathbf{x} \rangle \in [a, b]] = \frac{\mathbb{P}_{D^*}[\langle \mathbf{v}, \mathbf{x} \rangle \in [a, b] \text{ and } \langle \mathbf{w}, \mathbf{x} \rangle \in [-\sigma, \sigma]]}{\mathbb{P}_{D^*}[\langle \mathbf{w}, \mathbf{x} \rangle \in [-\sigma, \sigma]]}.$$

After projecting onto $\text{span}(\mathbf{v}, \mathbf{w})$ (an operation that preserves logconcavity), the numerator is the probability mass under a rectangle with side lengths $|b - a|$ and $2\sigma$, which is at most $O(\sigma|b - a|)$ as by Proposition A.2(b) the density is pointwise upper bounded by a constant. The claim follows since the denominator is $\Omega(\sigma)$.

Now we are ready to apply Proposition A.3. We see that if $D_{|T}$ matches moments of degree at most $k$ with $D^*_{|T}$ up to an additive slack of $d^{-O(k)}$, then $|\mathbb{E}_{D^*}[f \mid T] - \mathbb{E}_D[f \mid T]| \leq \widetilde{O}(1/\sqrt{k})$. Rewriting in terms of $\tau$ gives the theorem.

## B.2 PROOF OF PROPOSITION 3.2

The tester $T_1$ does the following:

1. For all $\alpha \in \mathbb{Z}^d_{\geq 0}$ with $|\alpha| = k$:
   (a) Compute the corresponding moment $\mathbb{E}_{(\mathbf{x}, y) \sim D} \mathbf{x}^\alpha := \frac{1}{|S|} \sum_{\mathbf{x} \in S} \mathbf{x}^\alpha$.
   (b) If $\left| \mathbb{E}_{(\mathbf{x}, y) \sim D}[\mathbf{x}^\alpha] - \mathbb{E}_{\mathbf{x} \sim D^*}[\mathbf{x}^\alpha] \right| > \frac{1}{d^k}$ then reject.
2. If all the checks above passed, accept.

First, we claim that for some absolute constant $C_1$, if the tester above accepts, we have $\mathbb{E}_{(\mathbf{x}, y) \sim D}[(\langle \mathbf{v}, \mathbf{x} \rangle)^k] \leq (C_1 k)^{k/2}$ for any $\mathbf{v} \in \mathbb{S}^{d-1}$. To show this, we first recall that by Proposition A.2(e) it is the case that $\mathbb{E}_{(\mathbf{x}, y) \sim D^*}[(\langle \mathbf{v}, \mathbf{x} \rangle)^k] \leq (K_3 k)^{k/2}$. But we have

$$\left| \mathbb{E}_{(\mathbf{x}, y) \sim D}[(\langle \mathbf{v}, \mathbf{x} \rangle)^k] - \mathbb{E}_{(\mathbf{x}, y) \sim D^*}[(\langle \mathbf{v}, \mathbf{x} \rangle)^k] \right| \leq \sum_{\alpha: |\alpha| = k} \left| \mathbb{E}_{(\mathbf{x}, y) \sim D}[\mathbf{x}^\alpha] - \mathbb{E}_{\mathbf{x} \sim D^*}[\mathbf{x}^\alpha] \right|$$

$$\leq d^k \cdot \max_{\alpha: |\alpha| = k} \left| \mathbb{E}_{(\mathbf{x}, y) \sim D}[\mathbf{x}^\alpha] - \mathbb{E}_{\mathbf{x} \sim D^*}[\mathbf{x}^\alpha] \right| \leq 1$$

Together with the bound $\mathbb{E}_{(\mathbf{x}, y) \sim D^*}[(\langle \mathbf{v}, \mathbf{x} \rangle)^k] \leq (K_3 k)^{k/2}$, the above implies that $\mathbb{E}_{(\mathbf{x}, y) \sim D}[(\langle \mathbf{v}, \mathbf{x} \rangle)^k] \leq (C_1 k)^{k/2}$ for some constant $C_1$.

Now, we need to show that if the elements of $S$ are chosen i.i.d. from $D^*$, and $|S| \geq \left( d^k, \left( \log \frac{1}{\delta} \right)^k \right)^{C_1}$ then the tester above accepts with probability at least $1 - \delta$. Consider any specific

multi-index $\alpha \in \mathbb{Z}_{\geq 0}^d$ with $|\alpha| = k$. Now, by Proposition A.2(f) we have the following:

$$\mathop{\mathbb{E}}_{\mathbf{x} \sim D^*}\left[\left(\mathbf{x}^\alpha - \mathop{\mathbb{E}}_{\mathbf{z} \sim D^*}[\mathbf{z}^\alpha]\right)^{2\log(1/\delta)}\right] \leq \sum_{\ell=0}^{2\log(1/\delta)} \left(\mathop{\mathbb{E}}_{\mathbf{x} \sim D^*}(\mathbf{x}^\alpha)^\ell\right) \cdot \left(\mathop{\mathbb{E}}_{\mathbf{z} \sim D^*}[\mathbf{z}^\alpha]\right)^{2\log(1/\delta)-\ell}$$

$$\leq \sum_{\ell=0}^{2\log(1/\delta)} (K_4 \ell k)^{\ell k/2}(K_4 k)^{k(2\log(1/\delta)-\ell)/2}$$

$$\leq 2\log(1/\delta)(2K_4\log(1/\delta)k)^{\log(1/\delta)k}$$

This, together with Markov's inequality implies that

$$\mathbb{P}\left[\left|\frac{1}{|S|}\sum_{\mathbf{x} \in S}\mathbf{x}^\alpha - \mathop{\mathbb{E}}_{\mathbf{x} \sim D^*}[\mathbf{x}^\alpha]\right| > \frac{1}{d^k}\right] \leq \left(\frac{d^k(3K_4 k\log(1/\delta))^{k/2}}{|S|}\right)^{2\log(1/\delta)}$$

Since $S$ is obtained by taking at least $|S| \geq \left(d^k, \left(\log\frac{1}{\delta}\right)^k\right)^{C_1}$, for sufficiently large $C_1$ we see that the above is upper-bounded by $\frac{1}{d^k}\delta$. Taking a union bound over all $\alpha \in \mathbb{Z}_{\geq 0}^d$ with $|\alpha| = k$, we see that with probability at least $1 - \delta$ the tester $T_1$ accepts, finishing the proof.

### B.3   PROOF OF PROPOSITION 3.3

Let $K_1$ be as in part (d) of Proposition A.2. The tester $T_2$ computes the fraction of elements in $S$ that are in $T$. If this fraction is $K_1\sigma/2$-close to $\mathbb{P}_{\mathbf{x} \sim D^*}[|\langle \mathbf{w}, \mathbf{x}\rangle| \leq \sigma]$, the algorithm accepts. The algorithm rejects otherwise.

Now, from (d) of Proposition A.2 we have that $\mathbb{P}_{\mathbf{x} \sim D^*}[|\langle \mathbf{w}, \mathbf{x}\rangle| \leq \sigma] \in [K_1\sigma, K_2\sigma]$. Therefore, if the fraction of elements in $S$ that belong in $T$ is $K_1\sigma/100$-close to $\mathbb{P}_{\mathbf{x} \sim D^*}[|\langle \mathbf{w}, \mathbf{x}\rangle| \leq \sigma]$, then this quantity is in $[K_1\sigma/2, (K_2 + K_1/2)\sigma]$ as required.

Finally, if $|S| \geq \frac{100}{K_1\sigma^2}\log\left(\frac{1}{\delta}\right)$ by standard Hoeffding bound, with probability at least $1 - \delta$ we indeed have that the fraction of elements in $S$ that are in $T$ is $K_1\sigma/2$-close to $\mathbb{P}_{\mathbf{x} \sim D^*}[|\langle \mathbf{w}, \mathbf{x}\rangle| \leq \sigma]$.

### B.4   PROOF OF PROPOSITION 3.4

The tester $T_3$ does the following:

1. Runs the tester $T_2$ from Proposition 3.3. If $T_2$ rejects, $T_3$ rejects as well.
2. Let $S_{|T}$ be the set of elements in $S$ for which $\mathbf{x} \in T$.
3. Let $k = \tilde{O}(1/\tau^2)$ be chosen as in Proposition 3.1.
4. For all $\alpha \in \mathbb{Z}_{\geq 0}^d$ with $|\alpha| = k$:
    (a) Compute the corresponding moment $\mathbb{E}_{(\mathbf{x},y) \sim D}[\mathbf{x}^\alpha \mid \mathbf{x} \in T] := \frac{1}{|S_{|T}|}\sum_{\mathbf{x} \in S_{|T}}\mathbf{x}^\alpha$.
    (b) If $\left|\mathbb{E}_{(\mathbf{x},y)\sim D}[\mathbf{x}^\alpha \mid \mathbf{x} \in T] - \mathbb{E}_{\mathbf{x} \sim D^*}[\mathbf{x}^\alpha \mid \mathbf{x} \in T]\right| > \frac{\tau}{d^k} \cdot d^{-\tilde{O}(k)}$ then reject, where the polylogarithmic in $d^{-\tilde{O}(k)}$ is chosen to satisfy the additive slack condition in Proposition 3.1.
5. If all the checks above passed, accept.

First, we argue that if the checks above pass, then Equations 3.3 and 3.4 will hold. If the tester passes, Equation 3.3 follows immediately from the guarantees in step (4b) of $T_3$ together with Proposition 3.1. Equation 3.4, in turn, is proven as follows:

$$\left|\mathop{\mathbb{E}}_{(\mathbf{x},y)\sim D}[(\langle \mathbf{v}, \mathbf{x}\rangle)^2] - \mathop{\mathbb{E}}_{(\mathbf{x},y)\sim D^*}[(\langle \mathbf{v}, \mathbf{x}\rangle)^2]\right| \leq \sum_{\alpha:|\alpha|=2}\left|\mathop{\mathbb{E}}_{(\mathbf{x},y)\sim D}[\mathbf{x}^\alpha] - \mathop{\mathbb{E}}_{\mathbf{x} \sim D^*}[\mathbf{x}^\alpha]\right|$$

$$\leq d^2 \cdot \max_{\alpha:|\alpha|=2}\left|\mathop{\mathbb{E}}_{(\mathbf{x},y)\sim D}[\mathbf{x}^\alpha] - \mathop{\mathbb{E}}_{\mathbf{x} \sim D^*}[\mathbf{x}^\alpha]\right| \leq \tau$$

Now, we need to show that if the elements of $S$ are chosen i.i.d. from $D^*$, and $|S| \geq ...$ then the tester above accepts with probability at least $1 - \delta$. Consider any specific mult-index $\alpha \in \mathbb{Z}_{\geq 0}^d$ with $|\alpha| = k$. Now, by Proposition A.2(f) we have for any positive integer $\ell$ the following:

$$\mathbb{E}_{\mathbf{x} \sim D^*} \left[ \left| (\mathbf{x}^\alpha)^\ell \right| \right] \leq (K_4 \ell k)^{k/2}$$

But by Proposition A.2(d) we have that $\mathbb{P}_{\mathbf{x} \sim D^*}[\mathbf{x} \in T] = \mathbb{P}_{\mathbf{x} \sim D^*}[|\langle \mathbf{x}, \mathbf{w} \rangle| \leq \sigma] \geq K_1 \sigma$. Therefore, the density of the distribution $D^*_{|T}$ (which is defined as the distribution one obtains by taking $D^*$ and conditioning on $\mathbf{x} \in T$) is upper bounded by the product of the density of the distribution $D^*$ and $\frac{1}{K_1 \sigma}$. This allows us to bound

$$\mathbb{E}_{\mathbf{x} \sim D^*} \left[ \left| (\mathbf{x}^\alpha)^\ell \right| \mid \mathbf{x} \in T \right] \leq \frac{1}{K_1 \sigma} \mathbb{E}_{\mathbf{x} \sim D^*} \left[ \left| (\mathbf{x}^\alpha)^\ell \right| \right] \leq \frac{(K_4 \ell k)^{k/2}}{K_1 \sigma}$$

This implies that

$$\mathbb{E}_{\mathbf{x} \sim D^*} \left[ \left( \mathbf{x}^\alpha - \mathbb{E}_{\mathbf{z} \sim D^*} [\mathbf{z}^\alpha \mid \mathbf{z} \in T] \right)^{2 \log(1/\delta)} \mid \mathbf{x} \in T \right]$$

$$\leq \sum_{\ell=0}^{2 \log(1/\delta)} \left( \mathbb{E}_{\mathbf{x} \sim D^*} \left[ (\mathbf{x}^\alpha)^\ell \mid \mathbf{x} \in T \right] \right) \cdot \left( \mathbb{E}_{\mathbf{x} \sim D^*} [(\mathbf{x}^\alpha \mid \mathbf{x} \in T]) \right)^{2 \log(1/\delta) - \ell}$$

$$\leq \frac{1}{(K_1 \sigma)^{2 \log(1/\delta)}} \sum_{\ell=0}^{2 \log(1/\delta)} (K_4 \ell k)^{\ell k/2} (K_4 k)^{k(2 \log(1/\delta) - \ell)/2}$$

$$\leq \frac{1}{(K_1 \sigma)^{2 \log(1/\delta)}} 2 \log(1/\delta) (2 K_4 \log(1/\delta) k)^{\log(1/\delta) k}$$

This, together with Markov's inequality implies that

$$\mathbb{P} \left[ \left| \frac{1}{|S|} \sum_{\mathbf{x} \in S} \mathbf{x}^\alpha - \mathbb{E}_{\mathbf{x} \sim D^*} [\mathbf{x}^\alpha] \right| > \frac{\tau}{d^k} \cdot d^{-\tilde{O}(k)} \right] \leq \left( \frac{d^{\tilde{O}(k)} (3 K_4 k \log(1/\delta))^{k/2}}{K_1 \sigma |S_{|T}| \tau} \right)^{2 \log(1/\delta)}$$

Now, recall that the tester $T_2$ in step (1) accepted, we have $|S_{|T}| \geq \frac{1}{C_2 \sigma} |S|$. Since $S$ is obtained by taking at least $|S| \geq \left( \frac{1}{\tau} \cdot \frac{1}{\sigma} \cdot d^{\frac{1}{\tau^2} \log^{C_5}\left(\frac{1}{\tau}\right)} \cdot \left( \log \frac{1}{\delta} \right)^{\frac{1}{\tau^2} \log^{C_5}\left(\frac{1}{\tau}\right)} \right)^{C_5}$, for sufficiently large $C_5$ we see that the expression above is upper-bounded by $\frac{1}{d^k} \delta$. Taking a union bound over all $\alpha \in \mathbb{Z}_{\geq 0}^d$ with $|\alpha| = k$, we see that with probability at least $1 - \delta$ the tester $T_3$ accepts, finishing the proof.

## C   PROOFS FROM SECTION 4

We first present the following Proposition, which ensures that we can form a loss function with certain desired properties.

**Proposition C.1.** *There are constants $c, c' > 0$, such that for any $\sigma > 0$, there exists a continuously differentiable function $\ell_\sigma : \mathbb{R} \to [0,1]$ with the following properties.*

1. *For any $t \in [-\sigma/6, \sigma/6]$, $\ell_\sigma(t) = \frac{1}{2} + \frac{t}{\sigma}$.*

2. *For any $t > \sigma/2$, $\ell_\sigma(t) = 1$ and for any $t < -\sigma/2$, $\ell_\sigma(t) = 0$.*

3. *For any $t \in \mathbb{R}$, $\ell'_\sigma(t) \in [0, c/\sigma]$, $\ell'_\sigma(t) = \ell'_\sigma(-t)$ and $|\ell''_\sigma(t)| \leq c'/\sigma^2$.*

*Proof.* We define $\ell_\sigma$ as follows.

$$\ell_\sigma(t) = \begin{cases} \frac{t}{\sigma} + \frac{1}{2}, & \text{if } |t| \leq \frac{\sigma}{6} \\ 1, & \text{if } t > \frac{\sigma}{2} \\ 0, & \text{if } t < \frac{-\sigma}{2} \\ \ell^+(t), t \in \left( \frac{\sigma}{6}, \frac{\sigma}{2} \right] \\ \ell^-(t), t \in \left[ -\frac{\sigma}{2}, -\frac{\sigma}{6} \right) \end{cases}$$

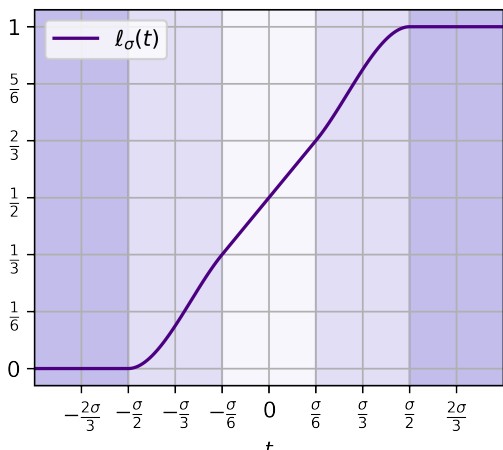

Figure 2: The function $\ell_\sigma$ used to smoothly approximate the ramp.

for some appropriate functions $\ell^+, \ell^-$. It is sufficient that we pick $\ell^+$ satisfying the following conditions (then $\ell^-$ would be defined symmetrically, i.e., $\ell^-(t) = 1 - \ell^+(-t)$).

- $\ell^+(\sigma/2) = 1$ and $\ell^{+\prime}(\sigma/2) = 0$.

- $\ell^+(\sigma/6) = 2/3$ and $\ell^{+\prime}(\sigma/6) = 1/\sigma$.

- $\ell^{+\prime\prime}$ is defined and bounded, except, possibly on $\sigma/6$ and/or $\sigma/2$.

We therefore need to satisfy four equations for $\ell^+$. So we set $\ell^+$ to be a degree 3 polynomial: $\ell^+(t) = a_1 t^3 + a_2 t^2 + a_3 t + a_4$. Whenever $\sigma > 0$, the system has a unique solution that satisfies the desired inequalities. In particular, we may solve the equation to get $a_1 = -9/\sigma^3, a_2 = 15/(2\sigma^2), a_3 = -3/(4\sigma)$ and $a_4 = 5/8$. For the resulting function (see Figure 2 below and Figure 4 in the appendix) we have that there are constants $c, c' > 0$ such that $\ell^{+\prime}(t) \in [0, c/\sigma]$ and $|\ell^{+\prime\prime}(t)| \leq c'/\sigma^2$ for any $t \in [\sigma/6, \sigma/2]$. $\qquad \square$

### C.1 PROOF OF LEMMA 4.3

We will prove the contrapositive of the claim, namely, that there are constants $c_1, c_2, c_3 > 0$ such that if $\angle(\mathbf{w}, \mathbf{w}^*), \angle(-\mathbf{w}, \mathbf{w}^*) > \frac{c_3}{\sqrt{1-2\eta}} \cdot \sigma$, and $\tau \leq c_2$, then $\|\nabla_{\mathbf{w}} \mathcal{L}_\sigma(\mathbf{w})\|_2 \geq c_1(1 - 2\eta)$.

Consider the case where $\angle(\mathbf{w}, \mathbf{w}^*) < \pi/2$ (otherwise, perform the same argument for $-\mathbf{w}$). Let $\mathbf{v}$ be a unit vector orthogonal to $\mathbf{w}$ that can be expressed as a linear combination of $\mathbf{w}$ and $\mathbf{w}^*$ and for which $\langle \mathbf{v}, \mathbf{w}^* \rangle = 0$. Then $\{\mathbf{v}, \mathbf{w}\}$ is an orthonormal basis for $V = \mathrm{span}(\mathbf{w}, \mathbf{w}^*)$. For any vector $\mathbf{x} \in \mathbb{R}^d$, we will use the following notation: $\mathbf{x_w} = \langle \mathbf{w}, \mathbf{x} \rangle$, $\mathbf{x_v} = \langle \mathbf{v}, \mathbf{x} \rangle$. It follows that $\mathrm{proj}_V(\mathbf{x}) = \mathbf{x_w} \mathbf{w} + \mathbf{x_v} \mathbf{v}$, where $\mathrm{proj}_V$ is the operator that orthogonally projects vectors on $V$.

Using the fact that $\nabla_{\mathbf{w}}(\langle \mathbf{w}, \mathbf{x} \rangle / \|\mathbf{w}\|_2) = \mathbf{x} - \langle \mathbf{w}, \mathbf{x} \rangle \mathbf{w} = \mathbf{x} - \mathbf{x_w} \mathbf{w}$ for any $\mathbf{w} \in \mathbb{S}^{d-1}$, the interchangeability of the gradient and expectation operators and the fact that $\ell'_\sigma$ is an even function we get that

$$\nabla_{\mathbf{w}} \mathcal{L}_\sigma(\mathbf{w}) = \mathbb{E}\left[ -\ell'_\sigma(|\langle \mathbf{w}, \mathbf{x} \rangle|) \cdot y \cdot (\mathbf{x} - \mathbf{x_w} \mathbf{w}) \right]$$

Since the projection operator $\mathrm{proj}_V$ is a contraction, we have $\|\nabla_{\mathbf{w}} \mathcal{L}_\sigma(\mathbf{w})\|_2 \geq \|\mathrm{proj}_V \nabla_{\mathbf{w}} \mathcal{L}_\sigma(\mathbf{w})\|_2$, and we can therefore restrict our attention to a simpler, two dimensional problem. In particular, since $\mathrm{proj}_V(\mathbf{x}) = \mathbf{x_w} \mathbf{w} + \mathbf{x_v} \mathbf{v}$, we get

$$\|\mathrm{proj}_V \nabla_{\mathbf{w}} \mathcal{L}_\sigma(\mathbf{w})\|_2 = \left\| \mathbb{E}\left[ -\ell'_\sigma(|\mathbf{x_w}|) \cdot y \cdot \mathbf{x_v} \mathbf{v} \right] \right\|_2 = \left| \mathbb{E}\left[ -\ell'_\sigma(|\mathbf{x_w}|) \cdot y \cdot \mathbf{x_v} \right] \right|$$

$$= \left| \mathbb{E}\left[ -\ell'_\sigma(|\mathbf{x_w}|) \cdot \mathrm{sign}(\langle \mathbf{w}^*, \mathbf{x} \rangle) \cdot (1 - 2 \, \mathbb{1}\{y \neq \mathrm{sign}(\langle \mathbf{w}^*, \mathbf{x} \rangle)\}) \cdot \mathbf{x_v} \right] \right|$$

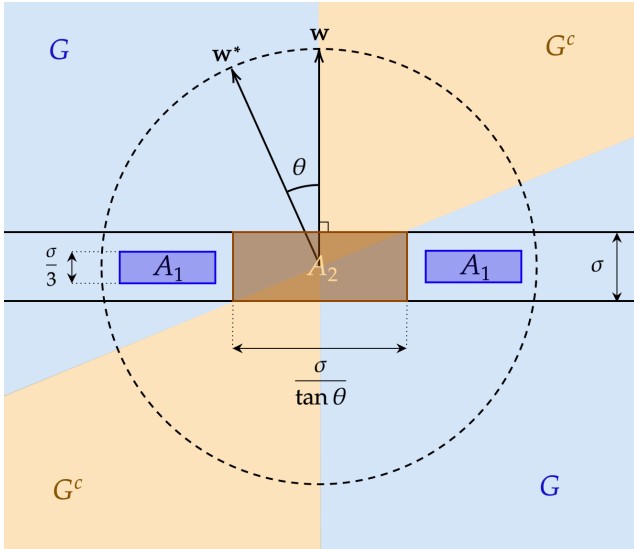

Figure 3: Critical regions in the proofs of main structural lemmas (Lemmas 4.3, 5.2). We analyze the contributions of the regions labeled $A_1, A_2$ to the quantities $A_1, A_2$ in the proofs. Specifically, the regions $A_1$ (which have height $\sigma/3$ so that the value of $\ell'_\sigma(\mathbf{x_w})$ for any $\mathbf{x}$ in these regions is exactly $1/\sigma$, by Proposition C.1) form a subset of the region $\mathcal{G}$, and their probability mass under $D_\mathcal{X}$ is (up to a multiplicative factor) a lower bound on the quantity $A_1$ (see Eq equation C.3). Similarly, the region $A_2$ is a subset of the intersection of $\mathcal{G}^c$ with the band of height $\sigma$, and has probability mass that is (up to a multiplicative factor) an upper bound on the quantity $A_2$ (see Eq equation C.4).

Let $F(y, \mathbf{x})$ denote $1 - 2\,\mathbb{1}\{y \neq \text{sign}(\langle \mathbf{w}^*, \mathbf{x}\rangle)\}$. We may write $\mathbf{x_v}$ as $|\mathbf{x_v}| \cdot \text{sign}(\mathbf{x_v})$ and let $\mathcal{G} \subseteq \mathbb{R}^2$ such that $\text{sign}(\mathbf{x_v}) \cdot \text{sign}(\langle \mathbf{w}^*, \mathbf{x}\rangle) = -1$ iff $\mathbf{x} \in \mathcal{G}$. Then, $\text{sign}(\mathbf{x_v}) \cdot \text{sign}(\langle \mathbf{w}^*, \mathbf{x}\rangle) = \mathbb{1}\{\mathbf{x} \notin \mathcal{G}\} - \mathbb{1}\{\mathbf{x} \in \mathcal{G}\}$. We get

$$\| \text{proj}_V \nabla_\mathbf{w} \mathcal{L}_\sigma(\mathbf{w})\|_2 =$$
$$= \left| \mathbb{E}\left[\ell'_\sigma(|\mathbf{x_w}|) \cdot (\mathbb{1}\{\mathbf{x} \in \mathcal{G}\} - \mathbb{1}\{\mathbf{x} \notin \mathcal{G}\}) \cdot F(y, \mathbf{x}) \cdot |\mathbf{x_v}|\right] \right| \geq$$
$$\geq \mathbb{E}\left[\ell'_\sigma(|\mathbf{x_w}|) \cdot \mathbb{1}\{\mathbf{x} \in \mathcal{G}\} \cdot F(y, \mathbf{x}) \cdot |\mathbf{x_v}|\right] - \mathbb{E}\left[\ell'_\sigma(|\mathbf{x_w}|) \cdot \mathbb{1}\{\mathbf{x} \notin \mathcal{G}\} \cdot F(y, \mathbf{x}) \cdot |\mathbf{x_v}|\right]$$

Let $A_1 = \mathbb{E}[\ell'_\sigma(|\mathbf{x_w}|) \cdot \mathbb{1}\{\mathbf{x} \in \mathcal{G}\} \cdot F(y, \mathbf{x}) \cdot |\mathbf{x_v}|]$ and $A_2 = \mathbb{E}[\ell'_\sigma(|\mathbf{x_w}|) \cdot \mathbb{1}\{\mathbf{x} \notin \mathcal{G}\} \cdot F(y, \mathbf{x}) \cdot |\mathbf{x_v}|]$. (See Figure 3.) Note that $\mathbb{E}_{y|\mathbf{x}}[F(y, \mathbf{x})] = 1 - 2\eta(\mathbf{x}) \in [1 - 2\eta, 1]$, where $1 - 2\eta > 0$. Therefore, we have that $A_1 \geq (1 - 2\eta) \cdot \mathbb{E}[\ell'_\sigma(|\mathbf{x_w}|) \cdot \mathbb{1}\{\mathbf{x} \in \mathcal{G}\} \cdot |\mathbf{x_v}|]$ and $A_2 \leq \mathbb{E}[\ell'_\sigma(|\mathbf{x_w}|) \cdot \mathbb{1}\{\mathbf{x} \notin \mathcal{G}\} \cdot |\mathbf{x_v}|]$.

Note that due to Proposition C.1, $\ell'_\sigma(|\mathbf{x_w}|) \leq c/\sigma$ for some constant $c$ and $\ell'_\sigma(|\mathbf{x_w}|) = 0$ whenever $|\mathbf{x_w}| > \sigma/2$. Therefore, if $\mathcal{U}_2$ is the band $B_\mathbf{w}(\sigma/2) = \{\mathbf{x} : |\mathbf{x_w}| \leq \sigma/2\}$ we have

$$A_2 \leq \frac{c}{\sigma} \cdot \mathbb{E}[\mathbb{1}\{\mathbf{x} \notin \mathcal{G}\} \cdot \mathbb{1}\{\mathbf{x} \in \mathcal{U}_2\} \cdot |\mathbf{x_v}|] \tag{C.1}$$

Moreover, for each individual $\mathbf{x}$, we have $\ell'_\sigma(|\mathbf{x_w}|) \cdot \mathbb{1}\{\mathbf{x} \in \mathcal{G}\} \cdot |\mathbf{x_v}| \geq 0$, due to the properties of $\ell'_\sigma$ (Proposition C.1). Hence, for any set $\mathcal{U}_1 \subseteq \mathbb{R}^d$ we have that

$$A_1 \geq (1 - 2\eta) \cdot \mathbb{E}[\ell'_\sigma(|\mathbf{x_w}|) \cdot \mathbb{1}\{\mathbf{x} \in \mathcal{G}\} \cdot \mathbb{1}\{\mathbf{x} \in \mathcal{U}_1\} \cdot |\mathbf{x_v}|]$$

Setting $\mathcal{U}_1 = B_\mathbf{w}(\sigma/6) = \{\mathbf{x} : |\mathbf{x_w}| \leq \sigma/6\}$, by Proposition C.1, we get $\ell'_\sigma(|\mathbf{x_w}|) \cdot \mathbb{1}\{\mathbf{x} \in \mathcal{U}_1\} = \frac{1}{\sigma} \cdot \mathbb{1}\{\mathbf{x} \in \mathcal{U}_1\}$.

$$A_1 \geq \frac{1 - 2\eta}{\sigma} \cdot \mathbb{E}[\mathbb{1}\{\mathbf{x} \in \mathcal{G}\} \cdot \mathbb{1}\{\mathbf{x} \in \mathcal{U}_1\} \cdot |\mathbf{x_v}|] \tag{C.2}$$

We now observe that by the definitions of $\mathcal{G}, \mathcal{U}_1, \mathcal{U}_2$, for any constant $R > 0$, there exist some constants $c', c'' > 0$ such that if $\sigma/\tan\theta < c'R$ (the points in $\mathbb{R}^2$ where $\partial\overline{\mathcal{G}}$ intersects either $\partial\mathcal{U}_1$ or

$\partial \mathcal{U}_2$ have projections on $\mathbf{v}$ that are $\Theta(\sigma/\tan\theta)$) we have that

$$\mathbb{1}\{\mathbf{x} \in \mathcal{G}\} \cdot \mathbb{1}\{\mathbf{x} \in \mathcal{U}_1\} \geq \mathbb{1}\{|\mathbf{x}_{\mathbf{v}}| \in [c'R, 2c'R]\} \cdot \mathbb{1}\{\mathbf{x} \in \mathcal{U}_1\} \quad \text{and}$$

$$\mathbb{1}\{\mathbf{x} \in \mathcal{G}\} \cdot \mathbb{1}\{\mathbf{x} \in \mathcal{U}_2\} \leq \mathbb{1}\{|\mathbf{x}_{\mathbf{v}}| \leq c''\sigma/\tan\theta\} \cdot \mathbb{1}\{\mathbf{x} \in \mathcal{U}_2\}$$

By equations equation C.1 and equation C.2, we get the following bounds whose graphical representations can be found in Figure 3.

$$A_1 \geq \frac{c'R(1-2\eta)}{\sigma} \cdot \mathbb{E}[\mathbb{1}\{|\mathbf{x}_{\mathbf{v}}| \in [c'R, 2c'R]\} \cdot \mathbb{1}\{\mathbf{x} \in \mathcal{U}_1\}] \tag{C.3}$$

$$A_2 \leq \frac{c \cdot c''}{\tan\theta} \cdot \mathbb{E}[\mathbb{1}\{|\mathbf{x}_{\mathbf{v}}| \leq c''\sigma/\tan\theta\} \cdot \mathbb{1}\{\mathbf{x} \in \mathcal{U}_2\}] \tag{C.4}$$

So far, we have used no distributional assumptions. Now, consider the corresponding expectations under the target marginal $D^*$ (which we assumed to be strongly log-concave).

$$I_1 = \mathop{\mathbb{E}}_{D^*}[\mathbb{1}\{|\mathbf{x}_{\mathbf{v}}| \in [c'R, 2c'R]\} \cdot \mathbb{1}\{\mathbf{x} \in \mathcal{U}_1\}]$$

$$I_2 = \mathop{\mathbb{E}}_{D^*}[\mathbb{1}\{|\mathbf{x}_{\mathbf{v}}| \leq c''\sigma/\tan\theta\} \cdot \mathbb{1}\{\mathbf{x} \in \mathcal{U}_2\}]$$

Any strongly log-concave distribution enjoys the "well-behaved" properties defined by (DKTZ20a), and therefore, if $R$ is picked to be small enough, then $I_1$ and $I_2$ are of order $\Theta(\sigma)$ (due to upper and lower bounds on the two dimensional marginal density over $V$ within constant radius balls – aka anti-anticoncentration and anticoncentration). Moreover, by Proposition A.2, we have $\mathbb{P}[\mathbf{x} \in \mathcal{U}_1]$ and $\mathbb{P}[\mathbf{x} \in \mathcal{U}_2]$ are both of order $\Theta(\sigma)$. Hence we have that there exist constants $c_1', c_2' > 0$ such that for the conditional expectations we have

$$\mathop{\mathbb{E}}_{D^*}\big[\, \mathbb{1}\{|\mathbf{x}_{\mathbf{v}}| \in [c'R, 2c'R]\} \,\big|\, \mathbb{1}\{\mathbf{x} \in \mathcal{U}_1\}\big] \geq c_1'$$

$$\mathop{\mathbb{E}}_{D^*}\big[\, \mathbb{1}\{|\mathbf{x}_{\mathbf{v}}| \leq c''\sigma/\tan\theta\} \,\big|\, \mathbb{1}\{\mathbf{x} \in \mathcal{U}_2\}\big] \leq c_2'$$

By assumption, Property equation 3.3 holds and, therefore, if $\tau \leq c_1'/2, c_2'/2 =: c_2$, we get that

$$\mathop{\mathbb{E}}_{D_{\mathcal{X}}}\big[\, \mathbb{1}\{|\mathbf{x}_{\mathbf{v}}| \in [c'R, 2c'R]\} \,\big|\, \mathbb{1}\{\mathbf{x} \in \mathcal{U}_1\}\big] \geq c_1'/2$$

$$\mathop{\mathbb{E}}_{D_{\mathcal{X}}}\big[\, \mathbb{1}\{|\mathbf{x}_{\mathbf{v}}| \leq c''\sigma/\tan\theta\} \,\big|\, \mathbb{1}\{\mathbf{x} \in \mathcal{U}_2\}\big] \leq c_2'/2$$

Moreover, by Property equation 3.2, we have that (under the true marginal) $\mathbb{P}[\mathbf{x} \in \mathcal{U}_1]$ and $\mathbb{P}[\mathbf{x} \in \mathcal{U}_2]$ are both $\Theta(\sigma)$. Hence, in total, we get that for some constants $\tilde{c}_1, \tilde{c}_2$, we have

$$A_1 \geq \tilde{c}_1 \cdot (1 - 2\eta) \text{ and } A_2 \leq \tilde{c}_2 \cdot \frac{\sigma}{\tan\theta}$$

Hence, if we pick $\sigma = \Theta((1-2\eta)\tan\theta)$, we get the desired result.

## C.2 PROOF OF PROPOSITION 4.4

For the following all the probabilities and expectations are over $D_{\mathcal{X}\mathcal{Y}}$. First we observe that

$$\mathbb{P}[y \neq \text{sign}(\langle\mathbf{w}, \mathbf{x}\rangle)] \leq \mathbb{P}[y \neq \text{sign}(\langle\mathbf{w}, \mathbf{x}\rangle) \cap y = \text{sign}(\langle\mathbf{w}^*, \mathbf{x}\rangle)] + \mathbb{P}[y \neq \text{sign}(\langle\mathbf{w}^*, \mathbf{x}\rangle)] \leq$$
$$\leq \mathbb{P}[\text{sign}(\langle\mathbf{w}, \mathbf{x}\rangle) \neq \text{sign}(\langle\mathbf{w}^*, \mathbf{x}\rangle)] + \mathsf{opt}.$$

Then, we observe that by assumption that $D_{\mathcal{X}\mathcal{Y}}$ satisfies Property equation 3.2, we have

$$\mathbb{P}[|\langle\mathbf{w}, \mathbf{x}\rangle| \leq \sigma] \leq C_3\sigma$$

and that

$$\mathbb{P}[\text{sign}(\langle\mathbf{w}, \mathbf{x}\rangle) \neq \text{sign}(\langle\mathbf{w}^*, \mathbf{x}\rangle) \cap |\langle\mathbf{w}, \mathbf{x}\rangle| > \sigma] \leq \mathbb{P}\Big[|\langle\mathbf{v}, \mathbf{x}\rangle| \geq \frac{\sigma}{\tan\theta}\Big],$$

where $\mathbf{v}$ is some vector perpendicular to $\mathbf{w}$. Using Markov's inequality, we get

$$\mathbb{P}\Big[|\langle\mathbf{v}, \mathbf{x}\rangle| \geq \frac{\sigma}{\tan\theta}\Big] \leq \frac{(\tan\theta)^k}{\sigma^k} \cdot \mathbb{E}[|\langle\mathbf{v}, \mathbf{x}\rangle|^k].$$

But, by assumption that $D_{\mathcal{X}\mathcal{Y}}$ satisfies Property equation 3.1, there is some constant $C_1 > 0$ such that $\mathbb{E}[|\langle \mathbf{v}, \mathbf{x} \rangle|^k] \leq (C_1 k)^{k/2}$. Thus

$$
\begin{aligned}
\mathbb{P}[\text{sign}(\langle \mathbf{w}, \mathbf{x} \rangle) \neq \text{sign}(\langle \mathbf{w}^*, \mathbf{x} \rangle)] &\leq \mathbb{P}[|\langle \mathbf{w}, \mathbf{x} \rangle| \leq \sigma] \\
&\quad + \mathbb{P}[\text{sign}(\langle \mathbf{w}, \mathbf{x} \rangle) \neq \text{sign}(\langle \mathbf{w}^*, \mathbf{x} \rangle) \cap |\langle \mathbf{w}, \mathbf{x} \rangle| > \sigma] \\
&\leq C_3 \sigma + \frac{(C_1 k)^{k/2} (\tan\theta)^k}{\sigma^k}.
\end{aligned}
$$

By picking $\sigma$ appropriately in order to balance the two terms (note that this is a different $\sigma$ than the one in Lemma 4.3), we get the desired result.

## D  PROOFS FROM SECTION 5

### D.1  PROOF OF THEOREM 5.1

We will follow the same steps as for proving Theorem 4.1. Once more, we draw a sufficiently large sample so that our testers are ensured to accept with high probability when the true marginal is indeed the target marginal $D^*$ and so that we have generalization, i.e. the guarantee that any approximate minimizer of the empirical error (error on the uniform empirical distribution over the sample drawn) is also an approximate minimizer of the true error. The algorithm we use is once more Algorithm 1, but this time we make multiple calls for different parameters $\sigma$ (and we run $T_1$ with higher $k$, as we will see shortly) and reject if any of these calls rejects. If we accept, we output the output of the execution of Algorithm 1 with the minimum empirical error.

The main difference between the Massart noise case and the agnostic case is that in the former we were able to pick $\sigma$ arbitrarily small, while in the latter we face a more delicate tradeoff. To balance competing contributions to the gradient norm, we must ensure that $\sigma$ is at least $\Theta(\text{opt})$ while also ensuring that it is not too large. And since we do not know the value of opt, we will need to search over a space of possible values for $\sigma$ that is only polynomially large in relevant parameters (similar to the approach of (DKTZ20b)). In our case, we may sparsify the space $(0, 1]$ of possible values for $\sigma$ up to accuracy $\Theta((\frac{\epsilon}{\sqrt{k}})^{1+1/k})$ and form a list of $\text{poly}(k/\epsilon)$ possible values for $\sigma$, one of which will satisfy $c_1\sigma - \Theta((\frac{\epsilon}{\sqrt{k}})^{1+1/k}) \leq \text{opt} \leq c_1\sigma$. hence, we perform the same (testing-learning) process for each of the possible values of $\sigma$ and get a list of candidate vectors which is still of polynomial size.

The final step is, again, to use Proposition 4.4, after running tester $T_1$ with parameter $k$ (Proposition 3.2) and tester $T_2$ with appropriate parameters for each of the candidate weight vectors. We get that our list contains a vector $\mathbf{w}$ with

$$
\mathbb{P}_{D_{\mathcal{X}\mathcal{Y}}}[y \neq \text{sign}(\langle \mathbf{w}, \mathbf{x} \rangle)] \leq \text{opt} + c \cdot k^{1/2} \cdot \theta^{1-1/(k+1)},
$$

where $\angle(\mathbf{w}, \mathbf{w}^*) \leq \theta := c_2\sigma$ for $\sigma$ such that $c_1\sigma - \Theta((\frac{\epsilon}{\sqrt{k}})^{1+1/k}) \leq \text{opt} \leq c_1\sigma$.

$$
\mathbb{P}_{D_{\mathcal{X}\mathcal{Y}}}[y \neq \text{sign}(\langle \mathbf{w}, \mathbf{x} \rangle)] \leq \text{opt} + c\sqrt{k} \cdot \left(\frac{c_2}{c_1}\text{opt} + \Theta\left(\left(\frac{\epsilon}{\sqrt{k}}\right)^{1+\frac{1}{k}}\right)\right)^{1-\frac{1}{k+1}} \leq O(\sqrt{k} \cdot \text{opt}^{1-\frac{1}{k+1}}) + \epsilon.
$$

However, we do not know which of the weight vectors in our list is the one guaranteed to achieve small error. In order to discover this vector, we estimate the probability of error of each of the corresponding halfspaces (which can be done efficiently, due to Hoeffding's bound) and pick the one with the smallest error. This final step does not require any distributional assumptions and we do not need to perform any further tests.

In order to obtain our $\tilde{O}(\text{opt})$ quasipolynomial time guarantee, observe first that we may assume without loss of generality that $\text{opt} \geq 1/d^C$ for some $C$; if instead $\text{opt} = o(1/d^2)$, say, then a sample of $O(d)$ points will with high probability be noiseless, and so simple linear programming will recover a consistent halfspace that will generalize. Moreover, we may assume that $\text{opt} \leq 1/10$, since otherwise achieving $O(\text{opt})$ is trivial (we may output an arbitrary halfspace). Let us adapt our algorithm so that we run tester $T_1$ (see Proposition 3.2) multiple times for all $k = 1, 2, \ldots, \lceil \log^2 d \rceil$ (this only changes our time and sample complexity by a $\text{polylog}(d)$ factor). Then Proposition 4.4

holds for some $k^*$ such that $k^* \in [\log(1/\mathsf{opt}), 2\log(1/\mathsf{opt})]$, since the interval has length at least $1$ (and therefore it contains some integer) and $2\log(1/\mathsf{opt}) \leq 2C\log d \leq \log^2 d$ (for large enough $d$). Therefore, by picking the best candidate we get a guarantee of order

$$
\begin{aligned}
\sqrt{k^*} \cdot \mathsf{opt}^{1-1/k^*} &= \sqrt{k^*} \cdot \mathsf{opt}^{-1/k^*} \mathsf{opt} \\
&= \sqrt{k^*} \cdot 2^{\frac{1}{k^*}\log\frac{1}{\mathsf{opt}}} \cdot \mathsf{opt} \\
&\leq \sqrt{2\log(1/\mathsf{opt})} \cdot 2 \cdot \mathsf{opt} \qquad \text{(since } \log(1/\mathsf{opt}) \leq k^* \leq 2\log(1/\mathsf{opt})) \\
&= \widetilde{O}(\mathsf{opt}) \,.
\end{aligned}
$$

This concludes the proof of Theorem 5.1.

### D.2 PROOF OF LEMMA 5.2

In the agnostic case, the proof is analogous to the proof of Lemma 4.3. However, in this case, the difference is that the random variable $F(y, \mathbf{x}) = 1 - 2\,\mathbb{1}\{y \neq \mathrm{sign}(\langle \mathbf{w}^*, \mathbf{x}\rangle)\}$ does not have conditional expectation on $\mathbf{x}$ that is lower bounded by a constant. Instead, we need to consider an additional term $A_3$ correcponding to the part $2\,\mathbb{1}\{y \neq \mathrm{sign}(\langle \mathbf{w}^*, \mathbf{x}\rangle)\}$ and the term $A_1$ will not be scaled by the factor $(1 - 2\eta)$ as in Lemma 4.3. Hence, with similar arguments we have that

$$
\|\nabla_{\mathbf{w}}\mathcal{L}_\sigma(\mathbf{w})\|_2 \geq A_1 - A_2 - A_3 \,,
$$

where $A_1 \geq \tilde{c}_1$, $A_2 \leq \tilde{c}_2 \cdot \frac{\sigma}{\tan\theta}$ and (using properties of $\ell'_\sigma$ as in Lemma 4.3 and the Cauchy-Schwarz inequality)

$$
\begin{aligned}
A_3 = 2\,\mathbb{E}[\ell'_\sigma(|\mathbf{x}_{\mathbf{w}}|) \cdot \mathbb{1}\{\mathbf{x} \in \mathcal{G}\} \cdot \mathbb{1}\{y \neq \mathrm{sign}(\langle \mathbf{w}, \mathbf{x}\rangle)\} \cdot |\mathbf{x}_{\mathbf{v}}|] &\leq \\
\leq \frac{2c}{\sigma} \cdot \mathbb{E}[\mathbb{1}\{\mathbf{x} \in \mathcal{U}_2\} \cdot \mathbb{1}\{y \neq \mathrm{sign}(\langle \mathbf{w}, \mathbf{x}\rangle)\} \cdot |\mathbf{x}_{\mathbf{v}}|] &\leq \\
\leq \frac{2c}{\sigma} \cdot \sqrt{\mathbb{E}[\mathbb{1}\{\mathbf{x} \in \mathcal{U}_2\} \cdot (\mathbf{x}_{\mathbf{v}})^2]} \cdot \sqrt{\mathbb{E}[\mathbb{1}\{y \neq \mathrm{sign}(\langle \mathbf{w}, \mathbf{x}\rangle)\}]} &= \\
= \frac{2c\sqrt{\mathsf{opt}}}{\sigma} \cdot \sqrt{\mathbb{E}[\langle \mathbf{v}, \mathbf{x}\rangle^2 \mid \mathbf{x} \in \mathcal{U}_2] \cdot \mathbb{P}[\mathbf{x} \in \mathcal{U}_2]} \,.
\end{aligned}
$$

Similarly to our approach in the proof of Lemma 4.3, we can use the assumed properties equation 3.2 and equation 3.4 to get that

$$
A_3 \leq \tilde{c}_3 \frac{\sqrt{\mathsf{opt}}}{\sqrt{\sigma}} \,,
$$

which gives that in order for the gradient loss to be small, we require $\mathsf{opt} \leq \Theta(\sigma)$.

### D.3 PROOF OF THEOREM 5.3

Before presenting the proof of Theorem 5.3, we prove the following Proposition, which is, essentially, a stronger version of Proposition 4.4 for the specific case when the target marginal distribution $D^*$ is the standard multivariate Gaussian distribution. Proposition D.1 is important to get an $O(\mathsf{opt})$ guarantee for the case where the target distribution is the standard Gaussian.

**Proposition D.1.** *Let $D_{\mathcal{X}\mathcal{Y}}$ be a distribution over $\mathbb{R}^d \times \{\pm 1\}$, $\mathbf{w}^* \in \arg\min_{\mathbf{w}\in\mathbb{S}^{d-1}} \mathbb{P}_{D_{\mathcal{X}\mathcal{Y}}}[y \neq \mathrm{sign}(\langle \mathbf{w}, \mathbf{x}\rangle)]$ and $\mathbf{w} \in \mathbb{S}^{d-1}$. Let $\theta \geq \angle(\mathbf{w}, \mathbf{w}^*)$ and suppose that $\theta \in [0, \pi/4]$. Then, for a sufficiently large constant $C$, there is a tester that given $\delta \in (0, 1)$, $\theta$, $\mathbf{w}$ and a set $S$ of samples from $D_{\mathcal{X}}$ with size at least $\left(\frac{d}{\theta}\log\frac{1}{\delta}\right)^C$, runs in time $\mathrm{poly}\left(\frac{1}{\theta}, d, \log\frac{1}{\delta}\right)$ and with probability $1 - \delta$ satisfies the following specifications:*

- *If the distribution $D_{\mathcal{X}}$ is $\mathcal{N}(0, I_d)$, the tester accepts.*

- *If the tester accepts, then we have:*

$$
\Pr_{\mathbf{x}\sim S}[\mathrm{sign}(\langle \mathbf{w}^*, \mathbf{x}\rangle) \neq \mathrm{sign}(\langle \mathbf{w}, \mathbf{x}\rangle)] \leq O(\theta)
$$

*Proof.* The testing algorithm does the following:

1. **Given:** Integer $d$, set $S \subset \mathbb{R}^d$, $\mathbf{w} \in \mathbb{S}^{d-1}$, $\theta \in (0, \pi/4]$ and $\delta \in (0,1)$.

2. Let $\text{proj}_{\perp \mathbf{w}} : \mathbb{R}^d \to \mathbb{R}^{d-1}$ denote the operator that projects a vector $\mathbf{x} \in \mathbb{R}^d$ to it's projection into the $(d-1)$-dimensional subspace of $\mathbb{R}^d$ that is orthogonal to $\mathbf{w}$.

3. For $i$ in $\left\{ 0, \pm 1, \cdots, \pm \frac{\sqrt{2 \log \frac{1}{\theta}}}{\theta} \right\}$

    (a) $S_i \leftarrow \{ \mathbf{x} \in S : \langle \mathbf{w}, \mathbf{x} \rangle \in [i\theta, (i+1)\theta] \}$

    (b) If $\frac{|S_i|}{|S|} > 2\theta$, then reject.

    (c) If $\left\| \frac{1}{|S_i|} \sum_{\mathbf{x} \in S_i} (\text{proj}_{\perp \mathbf{w}}(\mathbf{x}))(\text{proj}_{\perp \mathbf{w}}(\mathbf{x}))^T - I_{(d-1)} \right\|_{\text{op}} > 0.1$, reject.

4. If $\frac{1}{|S|} \sum_{\mathbf{x} \in S} \mathbb{1}_{|\langle \mathbf{w}, \mathbf{x} \rangle| > \sqrt{2 \log \frac{1}{\theta}}} > 5\theta$, then reject.

5. If reached this step, accept.

If the tester accepts, then we have the following properties for some sufficiently large constant $C' > 0$. For the following, consider the vector $\mathbf{v} \in \mathbb{R}^d$ to be the vector that is perpendicular to $\mathbf{w}$, lies within the plane defined by $\mathbf{w}$ and $\mathbf{w}^*$ and $\langle \mathbf{v}, \mathbf{w}^* \rangle \le 0$.

1. $\mathbb{P}_{\mathbf{x} \sim S}[|\langle \mathbf{w}, \mathbf{x} \rangle| \in [\theta i, \theta(i+1)]] \le C'\theta$, for any $i \in \left\{ 0, \pm 1, \ldots, \pm \frac{1}{\theta} \sqrt{2 \log \frac{1}{\theta}} \right\}$.

2. $\mathbb{P}_{\mathbf{x} \sim S_i}\left[ |\langle \mathbf{v}, \mathbf{x} \rangle| > \frac{\theta}{\tan \theta} \cdot i \right] \le C'/i^2$, for any $i \in \left\{ 0, \pm 1, \ldots, \pm \frac{1}{\theta} \sqrt{2 \log \frac{1}{\theta}} \right\}$.

3. $\mathbb{P}_{\mathbf{x} \sim S}\left[ |\langle \mathbf{w}, \mathbf{x} \rangle| \ge \sqrt{2 \log \frac{1}{\theta}} \right] \le C'\theta$.

Then, for $k = \frac{1}{\theta} \sqrt{2 \log \frac{1}{\theta}}$ and $\text{Strip}_i = \{ \mathbf{x} \in \mathbb{R}^d : \langle \mathbf{w}, \mathbf{x} \rangle| \in [\theta i, \theta(i+1)] \}$, we have that

$$\Pr_{\mathbf{x} \sim S}[\text{sign}(\langle \mathbf{w}, \mathbf{x} \rangle) \ne \text{sign}(\langle \mathbf{w}^*, \mathbf{x} \rangle)] \le$$

$$\sum_{i=-k}^{k} \mathbb{P}_{\mathbf{x} \sim S}[\mathbf{x} \in \text{Strip}_i] \cdot \mathbb{P}_{\mathbf{x} \sim S}\left[ |\langle \mathbf{v}, \mathbf{x} \rangle| > \frac{\theta}{\tan \theta} \cdot i \;\Big|\; \mathbf{x} \in \text{Strip}_i \right] + \mathbb{P}_{\mathbf{x} \sim S}\left[ |\langle \mathbf{w}, \mathbf{x} \rangle| \ge \sqrt{2 \log \frac{1}{\theta}} \right] \le$$

$$\sum_{i=-k}^{k} \frac{|S_i|}{|S|} \cdot \mathbb{P}_{\mathbf{x} \sim S_i}\left[ |\langle \mathbf{v}, \mathbf{x} \rangle| > \frac{\theta}{\tan \theta} \cdot i \right] + C'\theta \le (C')^2 \theta \cdot \left( 1 + \sum_{i \ne 0} \frac{2}{i^2} \right) + C'\theta = O(\theta)$$

Now, suppose the distribution $D_{\mathcal{X}}$ is indeed the standard Gaussian $\mathcal{N}(0, I_d)$. We would like to show that our tester accepts with probability at least $1 - \delta$. Since $D = \mathcal{N}(0, I_d)$, we see that for $\mathbf{x} \sim D$ we have that $\mathbf{x} \cdot \mathbf{w}$ is distributed as $\mathcal{N}(0,1)$. This implies that

- For all $i \in \left\{ 0, \pm 1, \cdots, \pm \frac{\sqrt{2 \log \frac{1}{\theta}}}{\theta} \right\}$ we have

    - $\Pr_{\mathbf{x} \sim \mathcal{N}(0, I_d)}[\langle \mathbf{w}, \mathbf{x} \rangle \in [i\theta, (i+1)\theta]] \le \frac{1}{\sqrt{2\pi}} \theta$

    - $\Pr_{\mathbf{x} \sim \mathcal{N}(0, I_d)}[\langle \mathbf{w}, \mathbf{x} \rangle \in [i\theta, (i+1)\theta]] \ge \theta \cdot \min_{x \in \left[ -\sqrt{2 \log \frac{1}{\theta}} - \theta, \sqrt{2 \log \frac{1}{\theta}} + \theta \right]} \frac{1}{\sqrt{2\pi}} e^{-\frac{x^2}{2}} \ge \frac{\theta^2}{10}$

- $\Pr_{\mathbf{x} \sim \mathcal{N}(0, I_d)}[\langle \mathbf{w}, \mathbf{x} \rangle \in [i\theta, (i+1)\theta]] \le \frac{1}{\sqrt{2\pi}} \theta$

- $\Pr_{\mathbf{x} \sim \mathcal{N}(0, I_d)}\left[ \langle \mathbf{w}, \mathbf{x} \rangle > 2\sqrt{\log \frac{1}{\theta}} \right] = \int_{2\sqrt{\log \frac{1}{\theta}}}^{\infty} \frac{1}{\sqrt{2\pi}} e^{-\frac{x^2}{2}} \, dx \le \theta \int_0^{\infty} \frac{1}{\sqrt{2\pi}} e^{-\frac{x^2}{2}} \, dx = \frac{\theta}{2}$

Therefore, via the standard Hoeffding bound, we see that for sufficiently large absolute constant $C$ we have with probability at least $1 - \frac{\delta}{4}$ over the choice of $S$ that

- For all $i \in \left\{ 0, \pm 1, \cdots, \pm \frac{\sqrt{2 \log \frac{1}{\theta}}}{\theta} \right\}$ we have

  - $\Pr_{\mathbf{x} \sim S} \left[ \langle \mathbf{w}, \mathbf{x} \rangle \in [i\theta, (i+1)\theta] \right] \leq \theta$
  - $\Pr_{\mathbf{x} \sim S} \left[ \langle \mathbf{w}, \mathbf{x} \rangle \in [i\theta, (i+1)\theta] \right] \geq \frac{\theta^2}{20}$

- $\Pr_{\mathbf{x} \sim S} \left[ \langle \mathbf{w}, \mathbf{x} \rangle > 2 \sqrt{\log \frac{1}{\theta}} \right] \leq \theta$

- $\Pr_{\mathbf{x} \sim S} \left[ \langle \mathbf{w}, \mathbf{x} \rangle < -2 \sqrt{\log \frac{1}{\theta}} \right] \leq \theta$

Finally, we would like to show that conditioned on the above, the probability of rejection in step (3b) is small.

**Fact D.2.** *Given a set $S \subset \mathbb{R}^{d-1}$ of i.i.d. samples from $\mathcal{N}(0, I_d)$, with probability at least $1 - \text{poly}\left( \frac{|S|}{d} \right)$ we have*

$$\left\| \frac{1}{|S|} \sum_{\mathbf{x} \in S} \mathbb{1}_{\langle \mathbf{w}, \mathbf{x} \rangle \in [i\theta, (i+1)\theta]} \mathbf{x}\mathbf{x}^T - I_{(d-1)} \right\|_{op} \leq 0.1$$

Now, since each sample $\mathbf{x}_i$ is drawn i.i.d. from $\mathcal{N}(0, I_d)$, we have that $\langle \mathbf{w}, \mathbf{x}_i \rangle$ and $\text{proj}_{\perp \mathbf{w}}(\mathbf{x}_i)$ are all independent from each other for all $i$. Since all the events we conditioned on depend on $\{\langle \mathbf{w}, \mathbf{x}_i \rangle\}$ we see that $\{\text{proj}_{\perp \mathbf{w}}(\mathbf{x}_i)\}$ are still distributed as i.i.d. samples from $\mathcal{N}(0, I_{(d-1)})$.

Recall that one of the events we have already conditioned on is that $\Pr_{\mathbf{x} \sim S} \left[ \langle \mathbf{w}, \mathbf{x} \rangle \in [i\theta, (i+1)\theta] \right] \geq \frac{\theta^2}{20}$ for all $i \in \left\{ 0, \pm 1, \cdots, \pm \frac{\sqrt{2 \log \frac{1}{\theta}}}{\theta} \right\}$. This allows us to lower bound by $\theta^2/20$ the ratio $|S_i|/|S|$. And since, as we described, for all these elements $\mathbf{x}_i$ the vectors $\text{proj}_{\perp \mathbf{w}}(\mathbf{x}_i)$ are distributed as i.i.d. samples from $\mathcal{N}(0, I_{(d-1)})$, we can use Fact D.2 to conclude that for sufficiently large absolute constant $C$, when $|S| = \left( \frac{d}{\theta} \log \frac{1}{\delta} \right)^C$ we have with probability $1 - \frac{\delta}{4}$ for all $i \in \left\{ 0, \pm 1, \cdots, \pm \frac{\sqrt{2 \log \frac{1}{\theta}}}{\theta} \right\}$ that

$$\left\| \frac{1}{|S_i|} \sum_{\mathbf{x} \in S_i} (\text{proj}_{\perp \mathbf{w}}(\mathbf{x}))(\text{proj}_{\perp \mathbf{w}}(\mathbf{x}))^T - I_{(d-1)} \right\|_{op} \leq 0.1$$

Overall, this allows us to conclude that with probability at least $1 - \delta$ the tester accepts. $\square$

We now present the proof of Theorem 5.3.

In the proof of Theorem 5.1, when the target distribution is the standard Gaussian in $d$ dimensions, we may apply Proposition D.1 (and run the corresponding tester), instead of Proposition 4.4, in order to ensure that our list will contain a vector $\mathbf{w}$ with

$$\mathbb{P}_{D_{\mathcal{X}\mathcal{Y}}} [y \neq \text{sign}(\langle \mathbf{w}, \mathbf{x} \rangle)] \leq \mathbb{P}_{D_{\mathcal{X}\mathcal{Y}}} [y \neq \text{sign}(\langle \mathbf{w}^*, \mathbf{x} \rangle)] + \mathbb{P}_{D_{\mathcal{X}\mathcal{Y}}} [\text{sign}(\langle \mathbf{w}^*, \mathbf{x} \rangle) \neq \text{sign}(\langle \mathbf{w}, \mathbf{x} \rangle)]$$

$$\leq \mathsf{opt} + O(\theta)$$

where $\angle(\mathbf{w}, \mathbf{w}^*) \leq \theta := c_2 \sigma$ and $\sigma$ is such that $c_1 \sigma - \Theta(\epsilon) \leq \mathsf{opt} \leq c_1 \sigma$, which gives the desired $O(\mathsf{opt}) + \epsilon$ bound. To get the value of $\sigma$ with the desired property, we once again sparsified the space $(0, 1]$ of possible values for $\sigma$, this time up to accuracy $\Theta(\epsilon)$.

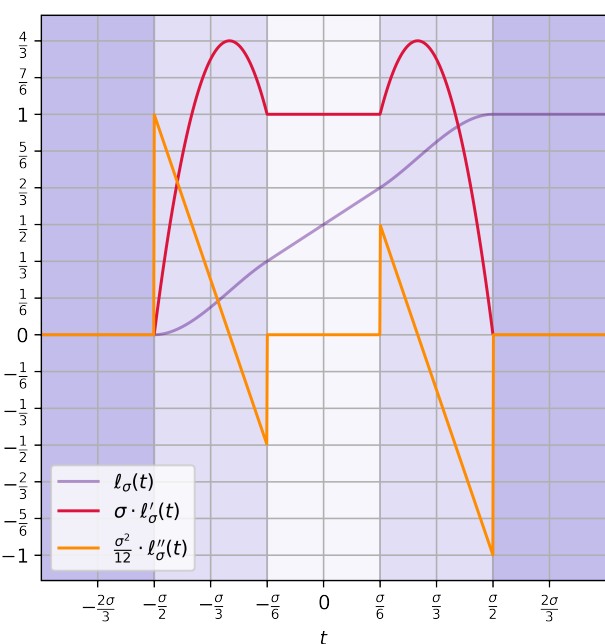

Figure 4: Figure illustrating the (normalized) first two derivatives of the function $\ell_\sigma$ used to define the non convex surrogate loss $\mathcal{L}_\sigma$. The normalization is appropriate since $\ell'_\sigma$ and $\ell''_\sigma$ are homogeneous in $1/\sigma$ and $1/\sigma^2$ respectively. In particular, we see that $\ell'_\sigma \leq \Theta(1/\sigma)$ and $|\ell''_\sigma| \leq \Theta(1/\sigma^2)$ everywhere.

