# OpenReview forum: "An Efficient Tester-Learner for Halfspaces"
_ICLR.cc/2024/Conference — ICLR 2024 poster_

### Official Review · Reviewer_9fsR · 2023-10-28

**Soundness:** 4 excellent
**Presentation:** 2 fair
**Contribution:** 3 good
**Rating:** 6
**Confidence:** 3

**Summary:**

In this paper, the authors give the first computationally efficient tester-learner for learning halfspaces where the target distribution is the d-variate Gaussian and the label noise is Massart or adversarial. The tester-learner framework was recently proposed as a generation to the distribution-specific learning setting where the algorithm needs to accept a dataset whenever it comes from a target distribution and needs to achieve the agnostic learning guarantee (error = $opt + \varepsilon$) whenever it accepts. Previous work only gave a sample-optimal tester learner for the same problem which was not computationally efficient. The main technical novelty in this work is tester that looks at the labels as opposed to the label-oblivious testers previously designed. For the adversarial noise setting, the authors achieve the suboptimal risk $O(opt) + \varepsilon$.

The authors build on the non-convex optimization approach of [DKTZ] which uses the a smoothed version of the ramp loss as a surrogate to the zero one loss. Although this is a non-convex function, it was shown that the stationary points are good solution which can be recovered by projected SGD as the first step under the Gaussianity assumption. In the testing-learning framework, we need to additionally check the following assumption: the probability masses of certain regions are proportional to their geometric measures. The tester checks local properties of the distribution in regions described by the stationary points using moment matching techniques. Naively, such a check could only guarantee the empirical mass is additively close to the true mass. However, using a refined moment test conditioned on a band based on the stationary vector (similar to the existing localization-based refinement techniques of Awasti et al. 2017) they could get the stronger multiplicative guarantee. This allows them to argue that if the test passes, the stationary points will indeed be close to the true weight vector in angular distance. This in turn means the returned vectors are good solutions using properties of Gaussian. The later step results in a larger error for the adversarial noise setting as opposed to the Massart noise setting.

**Strengths:**

The proposed work is an interesting combination of several technical ingredients that have been developed in learning theory for learning halfspaces and testing distributions such as non-convex optimization, fooling functions of halfspaces, and moment-matching tests. Moreover, they achieve the desired polynomial runtime for halfspaces in the newly proposed testing-learning framework.

**Weaknesses:**

The presentation could have been better. The paper has several forward references, that too from the main body to the appendix, which makes it slightly hard to follow.

**Questions:**

- Are the constants involved in the complexity very big? Given that ICLR accepts experiments and the dataset is easy to synthesize, how hard is it to implement and test the claimed efficient algorithm? This may be a general question targeted to even some of the prior works as well.
- I believe the results easily extend to non-homogeneous halfspaces where there is a constant offset term?
- I believe only the tester T3 uses the labels to check the fooling and T1 and T2 does not in Algo 1? Small typo: the Run T2 step has $\sigma$ missing in $B'_W(\sigma)$.

**Details Of Ethics Concerns:**

None.

---

> ### Author Response · Authors · 2023-11-22
>
> Thank you for your constructive comments and your time.
>
> **Q1:** The focus of our paper is theoretical and we did not optimize the constants or run experiments. Prior to our work, the best algorithm of [GKK ‘23] ran in time $d^{O(1/\epsilon^2)}$, which gets exponentially worse as $\epsilon$ decreases. In this work we show that a run-time of poly$(d, \epsilon)$ can be achieved laying the groundwork for practical algorithms for this task. We note that one difficulty with running experiments is that the guarantees in this paper (and the agnostic learning literature more broadly) hold in the worst case over all forms of adversarial label noise. When building a synthetic dataset, it is not obvious how to pick the noise in a way that exercises and demonstrates this worst-case guarantee.
>
> **Q2:** We believe that tester-learners which handle linear classifiers with non-zero offset terms may be significantly different from ours, given the fact that in standard distribution-specific agnostic learning the techniques used to handle halfspaces with non-zero offset terms are quite different from the techniques used here. See [Diakonikolas, Kontonis, Tzamos, Zarifis, ‘22] and [Diakonikolas, Kane, Stewart 2018] for further detail.
>
> **Q3:** The testers $T_1, T_2$ and $T_3$ do not use labels internally. However, testers $T_2$ and $T_3$ receive $\mathbf w$ as an input and $\mathbf w$ is computed using labelled examples (see Algorithm 1). Therefore, the tests we perform are indeed label-dependent, but the testers $T_1,T_2,T_3$ do not need to receive labelled examples per se. Additionally, thank you for pointing out the typo.
>
>
> *References:*
>
> Diakonikolas, I., Kontonis, V., Tzamos, C., & Zarifis, N. (2022, June). Learning general halfspaces with adversarial label noise via online gradient descent. In International Conference on Machine Learning (pp. 5118-5141). PMLR.
>
> Diakonikolas, I., Kane, D. M., & Stewart, A. (2018, June). Learning geometric concepts with nasty noise. In Proceedings of the 50th Annual ACM SIGACT Symposium on Theory of Computing (pp. 1061-1073).

---

### Official Review · Reviewer_CcgF · 2023-10-29

**Soundness:** 3 good
**Presentation:** 3 good
**Contribution:** 2 fair
**Rating:** 5
**Confidence:** 4

**Summary:**

This paper proposed a polynomial-time algorithm for learning halfspaces on testable fixed well-behaved distributions under Massart and adversarial noise. Unlike its prior works, it takes the labels into account and checks local properties of the distribution by testing the moments of the conditional distributions around the stationary points.

**Strengths:**

The framework of testable testing was recently proposed and has drawn great attention in the research community. This paper proposes a polynomial time algorithm for learning halfspaces under noisy settings, while the distributional assumptions are replaced by a tester on a fixed distribution. The paper is well-written. The technical parts look sound.

**Weaknesses:**

As mentioned in the paper, its subsequent work, “Tester-learners for halfspaces: Universal algorithms” has shown a more general tester-learner with stronger guarantees. This largely weakened the merit of publishing the work.

**Questions:**

Can you justify the unique value of this paper given the subsequent work has shown strictly stronger guarantees?

---

> ### Author Response · Authors · 2023-11-22
>
> We wish to thank the anonymous reviewer for their feedback. Please see our general response for a detailed comparison between our work and [GKSV '23].

---

### Official Review · Reviewer_HBcm · 2023-10-30

**Soundness:** 4 excellent
**Presentation:** 3 good
**Contribution:** 3 good
**Rating:** 8
**Confidence:** 3

**Summary:**

This paper worked on the problem of learning Gaussian Halfspace (with extension to more general stongly logconcave distributions) with Massart noise and agnostic noise, under the Tester-Learner models. The authors provided the first tester-learner algorithm with polynomial iteration and sample complexity, that achieves $\mathrm{OPT} + \epsilon$ error for the Massart noise and $O(\mathrm{OPT}) + \epsilon$ error for the agnostic noise (under Gaussian marginal). The technical contriutions of this paper are mainly the following: the authors devised more efficient testers using information of labels and exploiting the local geometric structure of the distribution (the condition probability on a band $P[v\cdot x \in [\alpha, \beta] | w\cdot x\in[-\sigma,\sigma]]$); they also showed that for some carefully designed loss function $\mathcal{L}_\sigma$, its stationary points $w$ are also vectors that are close (in angle) to the optimal solution $w^*$, under some distributions that are efficiently testable.

**Strengths:**

The paper contributes rigorously to the field of robustly learning halfspaces, and has some very interesting results.
1. Based on the results from DKTZ20a and GKK23, the authors devised new algorithms that are more efficient comparing to prior works. This includes a new loss function that work better for the specific task, and some new structrual results linking the gradient norm of this loss to the angle between the parameter $w$ and the optimal halfspace $w^*$.
2. The authors used some local property of the distribution that enables them to get desired result using testers that achieves only constant error rather than $\epsilon$ error.
3. These results can further extend from Gaussian distribution to strongly logconcave distribution, and get simialr results (at least for Massart noise).
4. The authors finally get the first polynomial tester-learner algorithm for learning Gaussian halfspaces under massart and agnostic noise.
5. The paper is clear and contains useful explanation on the intuiation of the algorithm.

**Weaknesses:**

I think there is no obvious weakness in general.

**Questions:**

1. I am confused why the algorithm needs two $T_3$ testers with different accuracies, $\sigma/6$ and $\sigma/2$?
2. I am not very familiar with tester-learner models. Are there lower bounds on learning gaussian halfspaces under massart/agnostic noise for tester-learner algorithms? Are tester-learner algorithms SQ algorithms?
3. In algorithm 1, what exactly is the function class $\mathcal{F}_{w'}$? How to choose the weights that are orthogonal to $w'$?

---

> ### Author Response · Authors · 2023-11-22
>
> We wish to thank the anonymous reviewer for their time and for appreciating of our work!
>
> **Q1:** Running tester $T_3$ with a given parameter $\sigma$ does not necessarily guarantee this tester accepts also for bigger parameter values $\sigma’$. Intuitively, this is because $\sigma$ denotes the width of a band around a given halfspace that the tester focuses on. Our analysis requires that the distribution is well-behaved for two different values of the band size $\sigma/2$ and $\sigma/6$, and therefore we run the tester $T_3$ for these two values separately.
>
> **Q2:** Since every testable learning algorithm will also satisfy the requirements for a distribution-specific agnostic learning, known hardness results for distribution-specific agnostic learning (such as [DK '22] and [DKMR '22]) also imply hardness of testable learning. We note that our tester-learner algorithms can be formulated in the SQ framework.
>
> **Q3:** The algorithm 1 does not compute function class $\mathcal F_{w}$, we only mention it as a comment for the reader to better understand what the various testers $T_1$, $T_2$ and $T_3$ accomplish. In particular, the class $\mathcal F_{w}$ is the class of functions that the marginal distribution (conditioned on the strip) has to fool in order to ensure soundness. For example, one function in $\mathcal F_{w}$ is $f$ such that $f(x) = 1$ for any $x\in A_2$ ($A_2$ as in Figure 1) and $f(x)=-1$ for any $x$ in the strip ($|x\cdot w| \le \sigma$), but $x$ not in $A_2$. Note that $f$ can be defined as an intersection of two halfspaces (orthogonal to $w$), since we only care about the values within the strip.

---

### Official Review · Reviewer_sjRf · 2023-10-31

**Soundness:** 4 excellent
**Presentation:** 4 excellent
**Contribution:** 3 good
**Rating:** 8
**Confidence:** 3

**Summary:**

This paper is a further exploration of the testable learning framework proposed by Rubinfeld and Vasilyan. The main feature of this framework is that it requires learning algorithms that learn near-optimal predictors whenever the input training sample passes a test (soundness), and also training samples pass the test whenever the distributional assumptions are met (completeness).

The paper gives polynomial-time testable learning algorithms for halfspaces when the marginal distribution is isotropic log-concave and:
(a) under Massart noise, guarantee error at most OPT + \epsilon. [Theorem 4.1]
(b). under adversarial noise, guarantee error at most O(OPT) + \epsilon. [Theorem 5.1]

**Strengths:**

This is a good paper that significantly extends what is known to be achievable in the testable-learning framework. To establish these results, the paper contributes new testing procedures that go beyond the limitations of prior work (Gollakota, Klivans, Kothari, 2023). Additionally, the techniques of this paper have also led to more general results in testable-learning (Gollakota, Klivans, Stavropoulos, Vasilyan, 2023).

The paper is well-written and easy to read. The authors do a great job discussing prior work, and how the paper fits with related literature.

**Weaknesses:**

The results may be a little limited in retrospect. In particular, the paper (Gollakota, Klivans, Stavropoulos, Vasilyan, NeurIPS 2023) already has more general results, including the results of this paper. If I understood correctly (based on page 2, subsequent work paragraph), there is some non-overlap in the techniques used in both papers, and so this paper may still be beneficial to the community.

**Questions:**

It would be great if the authors could discuss further the contributions of this paper in light of subsequent work (Gollakota, Klivans, Stavropoulos, Vasilyan, NeurIPS 2023). In particular, can the authors make a case for why the contributions in this paper are beneficial/useful given that more general results have already been published.

---

> ### Author Response · Authors · 2023-11-22
>
> Thank you for your comments and for appreciating our work! A more detailed comparison between our techniques and those in [GKSV '23] can be found in our global response.

---

> > ### Comment · Reviewer_sjRf · 2023-11-22
> >
> > Thank you for providing a detailed comparison with [GKSV '23]. It would be good to include this comparison/discussion in the paper. I have updated/increased my rating of the submission.

---

### Author Response · Authors · 2023-11-22

We wish to thank the anonymous reviewers for their constructive feedback.

Since two reviewers (sjRf and Ccgf) have asked about a more detailed comparison with [GKSV '23], we summarize the key value of this paper in this global response as follows:
- While it is true that [GKSV '23] achieves stronger results, this paper is the one that lays the fundamental groundwork for their approach, as they also acknowledge. Specifically, we are referring to the key framework of running SGD on a nonconvex surrogate and then running tests that estimate statistics of the conditional distribution restricted to a strip (where the strip itself is defined in a label-aware way in terms of an SGD stationary point). The contribution of [GKSV '23] is the extension of our tester to have a “universal” property, and the use of certifiable hypercontractivity in order to achieve this.
- In this paper, the tests are based on moment-matching, whereas [GKSV '23] uses a complicated semidefinite program resulting from a sum-of-squares relaxation. While such SDPs do technically run in polynomial time, they are notoriously slow in practice. In contrast, moment-matching is simple and easy to implement.
- From a conceptual point of view, this paper can also be seen as further investigating the power of moment matching as a primitive for testable learning. Since it is a fundamentally different type of test than say a sum-of-squares hypercontractivity test, it is subject to different (and potentially less restrictive) technical barriers. For example, the work of [GKSV '23] is only able to handle the class of general log-concave distributions under the assumption that they are certifiably hypercontractive (which is itself implied by the KLS conjecture). For our method, certifiable hypercontractivity is not a barrier and it is an interesting open question whether it can be extended to accept a target that is a general log-concave distribution without assuming the KLS conjecture.

---

### Meta-Review · Area_Chair_TPCo · 2023-12-05

**Metareview:**

The authors consider the testable learning model, where (1: soundness) the accuracy of the output hypothesis is near optimal when the training test passes the provided test, and (2: completeness) training sets drawn from a target distribution pass the test. They give the first computationally efficient algorithm for learning halfspaces in this model, achieving OPT$+\epsilon$ under Massart noise and any fixed strongly log-concave distribution and $O(\text{OPT})+\epsilon$ under adversarial noise and Gaussian distribution. The proofs are an interesting combination of techniques from learning theory including non-convex optimization and moment matching.

Reviewers agree the results are significant. The main shortcoming is that many of the results are superseded by a later paper (Gollakota, Klivans, Stavropoulos, Vasilyan, NeurIPS 2023). Given the fast pace of research in the field, I don't treat this as a reason to reject the paper.

**Justification For Why Not Higher Score:**

The later paper [GKSV23] gives stronger results.

**Justification For Why Not Lower Score:**

Reviewers liked the paper.

Given the fast pace of research in the field, I don't treat the later work as a reason to reject the paper.

---

### Decision · Program_Chairs · 2024-01-16

Accept (poster)